# Distinct signatures of calcium activity in brain mural cells

**Chaim Glück[1,2]\*, Kim David Ferrari[1,2], Noemi Binini[1,2], Annika Keller[2,3], Aiman S Saab[1,2], Jillian L Stobart[1,4], Bruno Weber[1,2]\***

[1]Institute of Pharmacology and Toxicology, University of Zurich, Zürich, Switzerland; [2]Neuroscience Center Zurich, University and ETH Zurich, Zurich, Switzerland; [3]Department of Neurosurgery, University of Zurich, Schlieren, Switzerland; [4]Rady Faculty of Health Sciences, College of Pharmacy, Winnipeg, Canada

**Abstract** Pericytes have been implicated in various neuropathologies, yet little is known about their function and signaling pathways in health. Here, we characterized calcium dynamics of cortical mural cells in anesthetized or awake *Pdgfrb*-CreERT2;Rosa26< LSL-GCaMP6s > mice and in acute brain slices. Smooth muscle cells (SMCs) and ensheathing pericytes (EPs), also named as terminal vascular SMCs, revealed similar calcium dynamics in vivo. In contrast, calcium signals in capillary pericytes (CPs) were irregular, higher in frequency, and occurred in cellular microdomains. In the absence of the vessel constricting agent U46619 in acute slices, SMCs and EPs revealed only sparse calcium signals, whereas CPs retained their spontaneous calcium activity. Interestingly, chemogenetic activation of neurons in vivo and acute elevations of extracellular potassium in brain slices strongly decreased calcium activity in CPs. We propose that neuronal activation and an extracellular increase in potassium suppress calcium activity in CPs, likely mediated by Kir2.2 and $K_{ATP}$ channels.

**\*For correspondence:**
chaim.glueck@uzh.ch (CG);
bweber@pharma.uzh.ch (BW)

**Competing interest:** The authors declare that no competing interests exist.

## Introduction

The entire abluminal surface of the cerebral vasculature is covered by mural cells, namely vascular smooth muscle cells (SMC) and pericytes, which exhibit a continuum of phenotypes. Despite several attempts to categorize these heterogeneous cells in the adult mouse by morphology and molecular footprints (*Hill et al., 2015*; *Hartmann et al., 2015a*; *Hartmann et al., 2015b*; *Vanlandewijck et al., 2018*), their identities remain unclear. In general, SMCs are ring-shaped cells, which express α-smooth muscle actin (αSMA) and surround arteries and penetrating arterioles. Pericytes are embedded within the vascular basement membrane and are classically described to have a protruding 'bump on a log' cell body with processes that run longitudinally along capillaries (*Attwell et al., 2016*). Many morphologically different subtypes of pericytes have been described (*Grant et al., 2019*; *Uemura et al., 2020*).

Pericytes regulate vascular morphogenesis and maturation as well as maintenance of the blood–brain barrier (*Armulik et al., 2011*; *Armulik et al., 2010*; *Daneman et al., 2010*). In the context of vascular development, several signaling pathways between endothelial cells (EC) and pericytes are important; for instance, in the recruitment of mural cells through platelet-derived growth factor B (PDGFB) and platelet-derived growth factor receptor beta (PDGFRβ) signaling (*Gaengel et al., 2009*), as well as Angiopoietin/Tie2 signaling in angiogenesis (*Teichert et al., 2017*). Furthermore, CNS pericytes in the mature cortex may provide a basal tone to the vasculature and contribute to vessel stability (*Berthiaume et al., 2018*). In recent years, pathologies such as diabetic retinopathy, Alzheimer's disease, circulatory failure, and primary familial brain calcification have been attributed to the loss or dysfunction of pericytes (*Kisler et al., 2017*; *Liu et al., 2019*; *Montagne et al., 2018*; *Winkler*

*et al., 2014*; *Zarb et al., 2019*; *Nikolakopoulou et al., 2019*). Yet, little is known about the function and signaling properties of pericytes in the healthy brain.

Being part of the neurovascular unit (NVU), SMCs and pericytes are in close contact with astrocytes, neurons, oligodendrocytes, and microglia. The NVU is involved in intricate regulatory mechanisms to tightly control blood flow (*Iadecola, 2017*). One of these mechanisms is functional hyperemia, which couples neural activity with changes in cerebral blood flow (CBF). Also, intrinsic vascular tone oscillations, known as vasomotion, have been observed maintaining blood 'flowmotion' in the brain at rest (*Intaglietta, 2017*). While it is evident that there is a relation between vasomotion and cytosolic calcium levels in SMCs (*Filosa et al., 2006*; *Longden et al., 2016*), participation of pericytes in vasomotion is currently debated (*Hill et al., 2015*; *Hall et al., 2014*; *Fernández-Klett et al., 2010*; *Peppiatt et al., 2006*). A recent study suggests that pericytes located at junctions of postarteriole transition regions are able to regulate blood flow through the capillary network (*Gonzales et al., 2020*). Moreover, specific optogenetic stimulation of capillary pericytes (CPs) could demonstrate a slow capillary constriction, suggesting that CPs might contribute to basal blood flow resistance (*Hartmann et al., 2021*).

Here, we combined in vivo and ex vivo approaches to investigate calcium signaling of mural cells in the somatosensory cortex vasculature of healthy adult mice. Given the distinct locations of morphologically diverse mural cells on the vasculature (*Hartmann et al., 2015a*; *Grant et al., 2019*), we wondered whether these cells would differ in their calcium signaling properties. Furthermore, we investigated how different stimuli such as vasomodulators and neuronal activity impact the calcium dynamics of CPs.

## Results

### Two-photon imaging of *Pdgfrb*-driven GCaMP6s in mural cells

To study calcium dynamics in mural cells, we crossed *Pdgfrb*-CreERT2 mice (*Gerl et al., 2015*) with Rosa26< LSL-GCaMP6s> (Ai96) reporter mice (*Figure 1A*). Measurements were performed in anesthetized and awake mice, as well as acute brain slices. For localization and classification of mural cells, we defined the vessel types based on their branch order and diameter. The continuum of mural cells along the arteriovenous axis was categorized, as described earlier (*Grant et al., 2019*), into SMC (0th branch), ensheathing pericytes (EP) (1st–4th branch), CP (>4th branch), and venular pericytes (VP) at postcapillary venules (*Figure 1B*, *Figure 1—figure supplement 1A-F*). The term EPs is used synonymous with precapillary or terminal vascular SMCs (*Hill et al., 2015*; *Hartmann et al., 2021*). Noteworthy is that in rare cases CPs interconnected vessel segments by extending processes through the parenchyma (*Figure 1—figure supplement 1D*). These few interconnecting pericytes were excluded from our analysis, since at the time it could not be determined, whether they are remnants of vessel regression or whether they represent a structural feature with importance in neurovascular coupling (*Brown, 2010*; *Alarcon-Martinez et al., 2020*; *Corliss et al., 2020*). Besides mural cells, some GCaMP6s-expressing astrocytes were sparsely detected (*Figure 1B*) and verified by labeling with astrocyte dye sulforhodamine 101 (SR101) (*Figure 1—figure supplement 2*). To avoid overlap of pericytic calcium signals by astrocytic signals, we omitted regions where a differentiation between GCaMP6s signals from astrocytes and pericytes was not possible.

### Distinct basal calcium transients of mural cells in vivo

In line with previous reports (*Hartmann et al., 2015a*; *Grant et al., 2019*), we found SMCs on surface arterioles and penetrating arterioles with an average inner diameter of 18.0 ± 7.3 µm (values are mean ± SD) (*Figure 2—figure supplement 1*). EPs were located on vessels of 1st–4th branch order with an average inner diameter of 7.3 ± 1.7 µm. CPs were found on capillaries (>4th branch order) with an average diameter of 4.1 ± 0.7 µm (*Figure 2—figure supplement 1*).

In lightly anesthetized animals (1.2% isoflurane), all the mural cells described above revealed basal calcium fluctuations in somata and processes (*Figure 2A–D*, *Video 1*, *Video 2*, *Video 3*, *Video 4*). Initial visual inspections of the calcium traces from SMCs and EPs indicated similar calcium dynamics as both revealed synchronous calcium signals (*Figure 2A and B*). Contrarily, CPs and VPs exhibited asynchronous calcium signals that are also appearing in microdomains along the processes (*Figure 2C and D*).

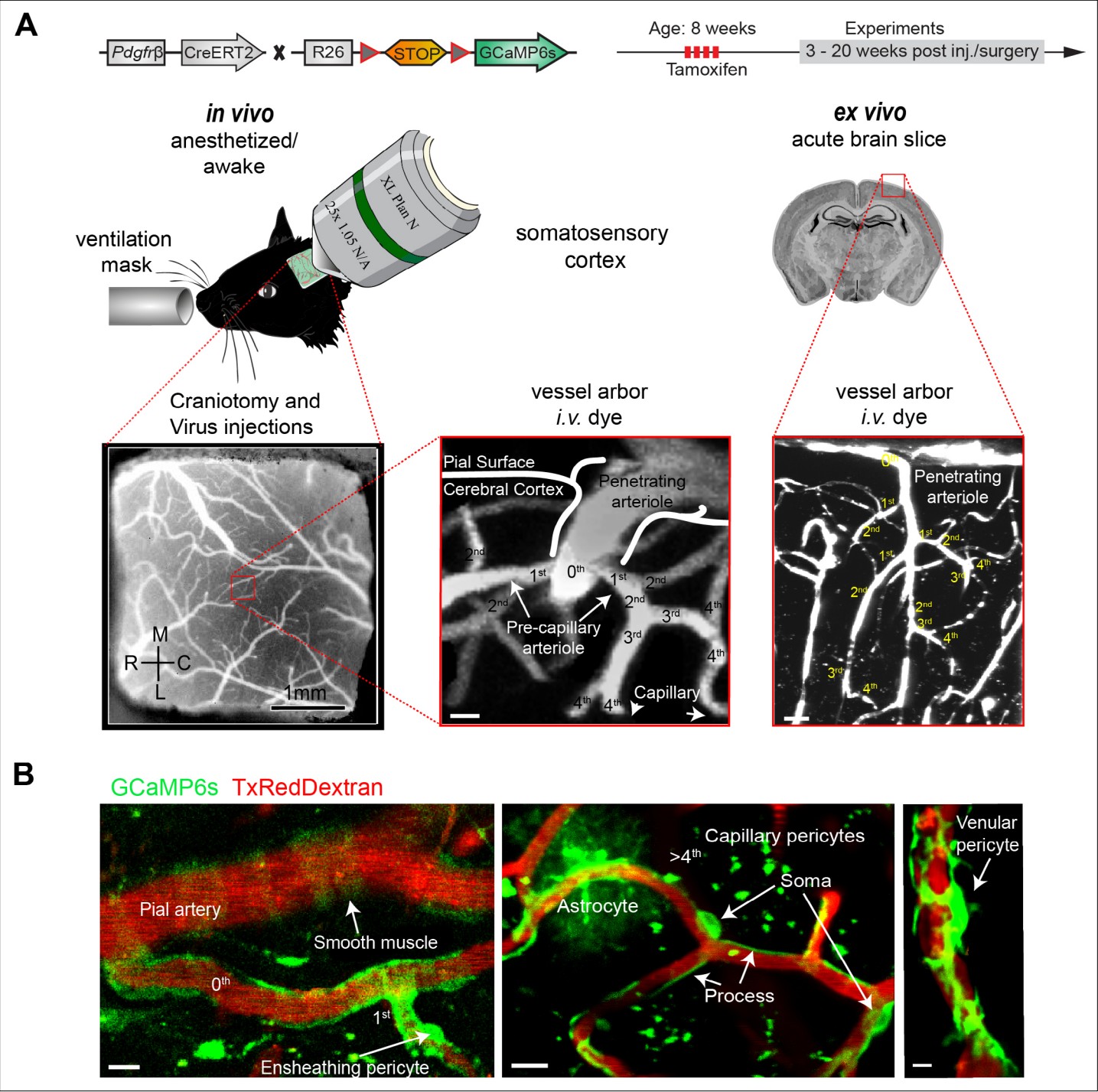

**Figure 1.** Two-photon calcium imaging of mural cells. (**A**) GCaMP6s expression in *Pdgfrb*-positive cells of *Pdgfrb*-CreERT2:R26-GCaMP6s[f/stop/f] transgenic mice, induced by four consecutive Tamoxifen injections. If required, adeno-associated viruses (AAV) were injected before the chronic cranial window implantation over the somatosensory cortex. Images were acquired at a wavelength of 940 nm, and the vasculature was labeled via an intravenous injection (iv.) of 2.5% Texas Red Dextran (70 kDa). In vivo experiments were conducted with awake and anesthetized (1.2% isoflurane, supplied by a ventilation mask) mice. For pharmacologic interventions, acute brain slices of the same mice were prepared. Z-stacks of the vessel arbor were acquired to determine the precise location of the imaged cells along the vasculature. (**B**) In vivo images of the cortical vasculature showing GCaMP6s (green) labeled mural cells. SMCs are located on pial arteries and ensheathing pericytes on 1st–4th branch order vessels. CPs (consisting of mesh and thin stranded pericytes) are found on vessels of >4th branch order. Occasionally, some astrocytes showed GCaMP6s expression. VPs reside at postcapillary venules. Scale bars: 10 µm. See also *Figure 1—figure supplements 1 and 2*.

The online version of this article includes the following figure supplement(s) for figure 1:

*Figure 1 continued on next page*

*Figure 1 continued*

**Figure supplement 1.** GCaMP6s expression in mural cells of *Pdgfrb*-CreERT2:R26-GCaMP6s$^{f/stop/f}$ mice.

**Figure supplement 2.** GCaMP6s expression in *Pdgfrb*-positive astrocytes of *Pdgfrb*-CreERT2:R26-GCaMP6s$^{f/stop/f}$ mice.

Calcium signals in SMCs during spontaneous vasomotion were inversely correlated with vessel diameter (transformed $r_{Fisher\ z}$ = 1.20 ± 0.37, n = 13, *Figure 2E and H*), as reported in a previous study (*Hill et al., 2015*). A similar behaviour of calcium signaling was observed in EPs (transformed $r_{Fisher\ z}$ = 1.06 ± 0.22, n = 13, *Figure 2F and H*). Moreover, power spectral analysis of the EP calcium signals showed a distinct peak at 0.1 Hz (*Figure 2—figure supplement 2*), which is in line with the vasomotion frequency observed in SMCs (*van Veluw et al., 2020*; *Mateo et al., 2017*). In accordance with earlier studies (*Pabelick et al., 2001*), we found that the oscillations in vessel diameter follow the calcium transients in EPs by an average delay of 300 ms, suggesting a calcium-dependent contraction mechanism. In contrast, there was no relationship between spontaneous calcium signals in CPs and capillary diameter (transformed $r_{Fisher\ z}$ = –0.005 ± 0.06, n = 9, *Figure 2G and H*).

To compare the calcium dynamics between different mural cells, we measured frequency, amplitude, and duration of calcium transients using semi-automated image analysis as described and employed earlier by our lab (*Zuend et al., 2020*; *Stobart et al., 2018b*). Regions of interest (ROI) for pericyte somata were selected by hand and ROIs for pericyte processes were automatically detected with an unbiased algorithm (*Figure 2—figure supplement 3*; *Ellefsen et al., 2014*) implemented in a custom-written MATLAB toolbox, CHIPS (*Barrett et al., 2018*).

In anesthetized animals, calcium signals of SMCs and EP somata were similar in amplitude and duration (*Figure 2J and K*); however, signal frequency (expressed as mean ± SD signals/min) was higher in EPs compared to SMCs (8.3 ± 2.8 vs. 7.1 ± 1.6, *Figure 2I*, *Table 1*). On the other hand, differences in calcium dynamics were more pronounced between EPs and CPs. Compared to EPs, calcium signals in CP processes were two times more frequent and about 30% shorter in duration (*Figure 2I and K*), whereas calcium signals in CP somata were almost twofold larger in amplitude and ~20% shorter in duration (*Figure 2J and K*, *Table 1*).

Prolonged isoflurane anesthesia has been shown to affect the cerebral vasculature and cause a decrease in vasomotor activity of arteries and arterioles (*van Veluw et al., 2020*; *Slupe and Kirsch, 2018*). We therefore only included measurements of SMCs and EPs up to 25 minutes post-anesthesia induction for analysis. Longer anesthesia (>30 min) led to a stark reduction in calcium activity (data not shown). However, to avoid possible anesthesia-related alterations on basal calcium activity of mural cells, we also performed calcium imaging in awake mice, as previously employed in our lab (*Zuend et al., 2020*; *Stobart et al., 2018a*). In the following we refer to anesthetized measurements as in vivo$_{anest}$ and awake measurements as in vivo$_{awake}$.

Indeed, there were differences in mural cell calcium dynamics between measurements in awake and anesthetized mice (*Figure 2—figure supplement 4*, *Table 1*). SMC calcium signals were more frequent in awake mice compared to anesthetized mice, while the calcium signal frequency in EPs and CPs was not significantly affected (*Figure 2—figure supplement 4A, D, G*, *Table 1*). Calcium signal amplitudes (expressed as df/f ± SD) in CPs were two to three times lower in awake compared to anesthetized animals, in both somata (in vivo$_{awake}$: 0.3 ± 0.1 vs. in vivo$_{anest.}$: 1.1 ± 1.3) and processes (in vivo$_{awake}$: 0.7 ± 0.3 vs. in vivo$_{anest.}$: 1.4 ± 0.4, *Figure 2—figure supplement 4H*, *Table 1*).

Importantly, the overall differences in calcium signaling signatures between mural cells that were observed in anesthetized mice (*Figure 2I–K*) were similar to those measured in awake mice (*Figure 2L–N*), emphasizing that more distal pericytes are functionally distinct. For example, signal frequency was two times higher in CP processes compared to EP processes (39.6 ± 14.5 vs. 20.2 ± 7.8, *Figure 2L*, *Table 1*). Also, signal amplitudes and durations differed significantly between CPs and EPs in both their somata and processes (*Figure 2M and N*, *Table 1*). Thus, EPs and CPs have distinct basal calcium dynamics, which, besides their location in the vascular network, morphology, and αSMA expression, can be used as a further measure to differentiate between these mural cell subtypes. Furthermore, despite transcriptional similarities between EPs and SMCs, subdomain (soma/process) calcium activity in EPs is a characteristic, which is not present in SMCs.

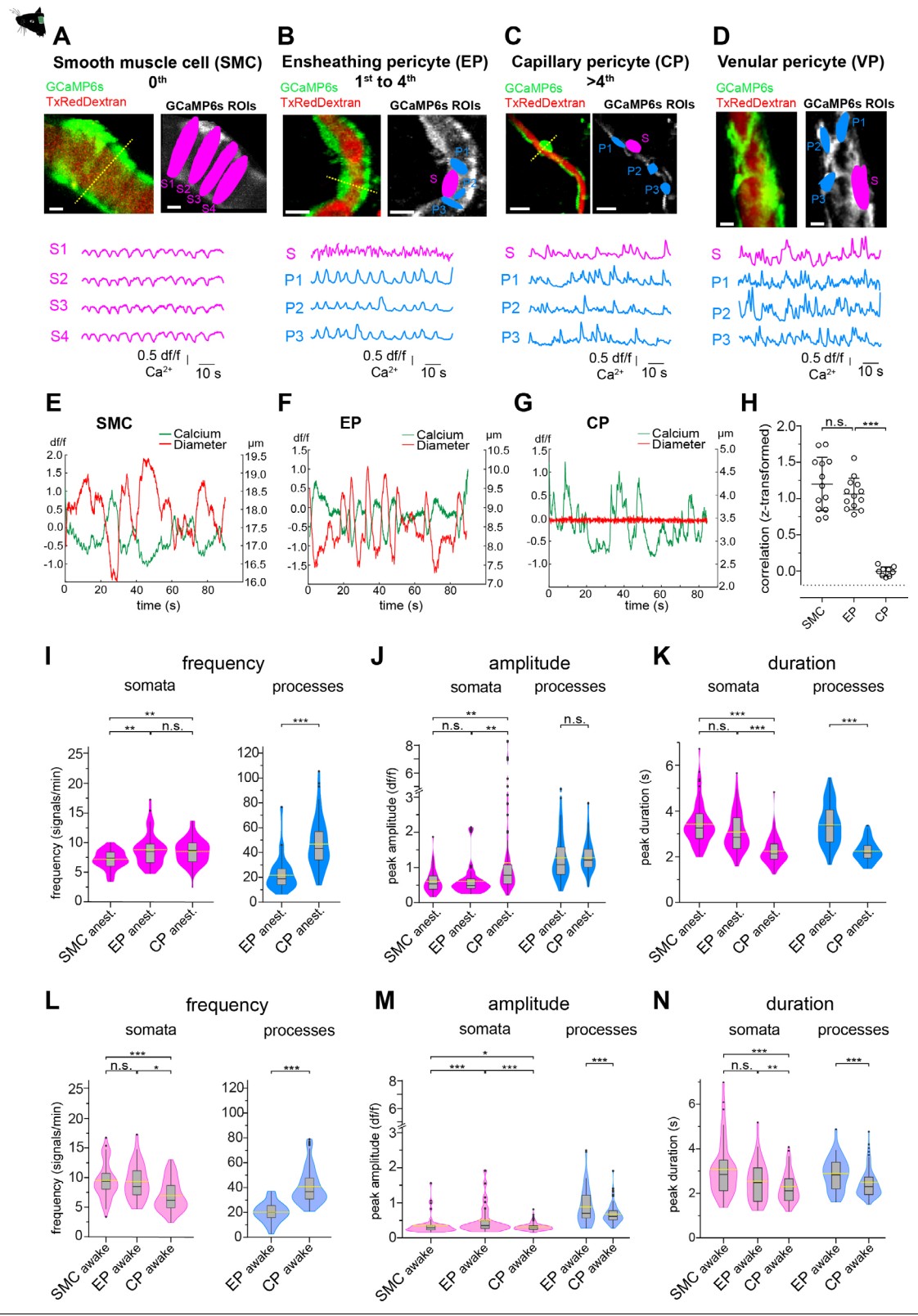

**Figure 2.** Mural cell calcium dynamics in vivo. (**A–D**) Representative images of (**A**) SMCs, (**B**) EP, (**C**) CP, and (**D**) VP in vivo. Regions of interest (ROIs) for somata (S, magenta) were hand selected. ROIs for processes (P1–3, blue) were found in an unbiased way, employing a MATLAB-based algorithm (see *Figure 2—figure supplement 3*). Below the images are the normalized time traces of calcium signals of GCaMP6s fluorescence extracted from the corresponding example ROIs. Vessel diameters were measured with line scans across somata (indicated by the dashed yellow lines). Scale bars: 10 µm.

*Figure 2 continued on next page*

*Figure 2 continued*

(**E–G**) Plots depicting the relation between cytosolic calcium (green trace) and vessel diameter (red trace) measured simultaneously with line scans along the soma, for (**E**) SMC, (**F**) EP, and (**G**) CP. Traces were centered on the average diameter of the vessel. (**H**) Comparison of the correlations (Fisher z-transformed) between calcium signals and changes in vessel diameter. Unpaired two-tailed t-tests were performed to compare groups. Individual values with mean ± SD are shown. SMC: n = 13, EP: n = 13, CP: n = 9. t(24) = 1.139, p=0.27; t(20) = 14.21, ***p<0.001. (**I–K**) Quantification and comparison of baseline calcium signal properties: (**I**) frequency (signals per minute), (**J**) peak amplitude (df/f), and (**K**) duration (s) between SMC, EP, and CP somata and processes of in vivo$_{anest}$ mice, shown as violin/box plots (Tukey whiskers). The dashed yellow lines indicate the mean values of the respective parameters. Statistics were calculated using linear mixed-effects models and Tukey post hoc tests, SMC$_{anest}$: N = 14, n = 67, EP$_{anest}$: N = 16, n = 53, CP$_{anest}$: N = 25, n = 93. *p<0.05, **p<0.01, ***p<0.001. (**L–N**) Quantification and comparison of baseline calcium signal properties: (**L**) frequency (signals per minute), (**M**) peak amplitude (df/f), and (**N**) duration (s) between SMC, EP, and CP somata and processes of in vivo$_{awake}$ mice, shown as violin/box plots (Tukey whiskers). The dashed yellow lines indicate the mean values of the respective parameters. Statistics were calculated using linear mixed-effects models and Tukey post hoc tests, SMC$_{awake}$: N = 4, n = 53; EP$_{awake}$: N = 3, n = 32; CP$_{awake}$: N = 3, n = 96. *p<0.05, **p<0.01, ***p<0.001. n.s. indicates not significant. See also *Figure 2—figure supplements 1–4*. Data: Table 1 and *Figure 2—source data 1*.

The online version of this article includes the following source data and figure supplement(s) for figure 2:

**Source data 1.** Source data for *Figure 2*.

**Figure supplement 1.** Vascular diameters associated with mural cells in awake and anesthetized mice.

**Figure supplement 1—source data 1.** Source data for *Figure 2—figure supplement 1*.

**Figure supplement 2.** Periodogram of EPs.

**Figure supplement 3.** Automatic ROI detection.

**Figure supplement 4.** Comparison of calcium signal properties of mural cells in awake vs. anesthetized animals.

**Figure supplement 4—source data 1.** Source data for *Figure 2—figure supplement 4*.

## Persisting calcium signals in CPs ex vivo

To further investigate calcium signal properties of mural cells, we prepared acute brain slices for ex vivo pharmacological probing. Blood plasma was stained with an iv. injection of 70 kDa Texas Red Dextran (100 µL, 2.5%) via the tail vein prior to the slice preparation. The fluorophore remained in the vasculature for several hours for easy capillary identification and further served as a control for motion artifacts. Orientation in the slice was obtained by following a penetrating artery from the slice surface to its branches (*Figure 1A*).

Interestingly, CPs in acute brain slices retained their highly frequent calcium transients, which were detected in both their somata and processes (*Figure 3A*). And CP spontaneous calcium signal frequency in both somata and processes in slices was not significantly different to in vivo$_{awake}$ and in vivo$_{anest}$ basal calcium activity (*Figure 3B*). However, signal amplitudes in CPs were on average larger in slices compared to in vivo$_{awake}$ and in vivo$_{anest}$ (*Figure 3—figure supplement 1*).

Surprisingly, in contrast to CPs, calcium activity of SMCs and EPs was greatly reduced ex vivo (*Figure 3C and D*). Compared to in vivo$_{anest}$, the calcium signal frequency of EPs was reduced more than fourfold in somata (ex vivo: 1.9 ± 1.7 vs. in vivo$_{anest}$.: 8.3 ± 2.8) and more than 10-fold in processes (ex vivo: 1.9 ± 2.2 vs. in vivo$_{anest}$: 21.6 ± 12.0). Similarly, in SMCs, the calcium signal frequency was reduced more than fourfold ex vivo compared to in vivo$_{anest}$ (1.6 ± 1.3 vs. 7.1 ± 1.6). We wondered whether the pronounced reduction of calcium activity in SMCs and EPs is linked to a loss of a vascular tone in slices. Most studies on mural cells in acute slice preparations include the addition of a thromboxane A$_2$ receptor agonist (U46619) to generate an artificial vascular tone (*Mishra*

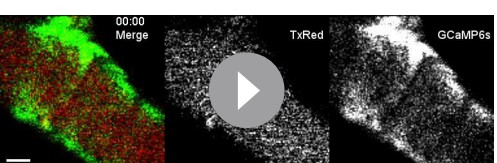

**Video 1.** Basal SMC calcium activity in vivo. Recorded at 11.84 Hz and played at 45 fps, showing Merge, TxRed, and GCaMP6s channels. Scale bar 5 µm.
https://elifesciences.org/articles/70591/figures#video1

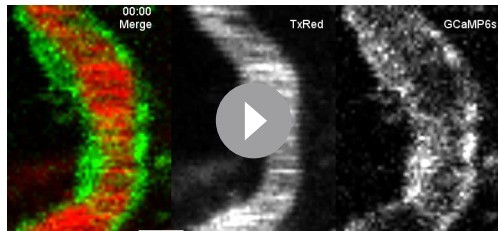

**Video 2.** Basal ensheathing pericyte calcium activity in vivo. Recorded at 11.84 Hz and played at 45 fps, showing Merge, TxRed, and GCaMP6s channels. Scale bar 10 µm.
https://elifesciences.org/articles/70591/figures#video2

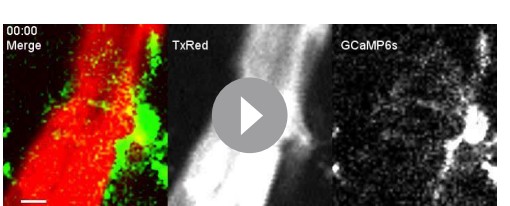

**Video 3.** Basal capillary pericyte calcium activity in vivo. Recorded at 11.84 Hz and played at 45 fps, showing Merge, TxRed, and GCaMP6s channels. Scale bar 10 μm.

https://elifesciences.org/articles/70591/figures#video3

*et al., 2014*; *Brown et al., 2002*). Strikingly, upon bath-application of U46619 (100 nM), calcium signal frequency in SMCs and EPs (both in somata and processes) increased more than fourfold (*Figure 3E*). Incidentally, a similar effect of U46619 on spontaneous activity in SMCs and proximal pericytes was recently reported in retinal preparations (*Gonzales et al., 2020*).

Curiously, in CPs, we observed that short pulses (1 min) of U46619 (100 nM) elicited a strong calcium response (*Figure 3G and H* and *Video 5*). This overt calcium response in CPs was accompanied by morphological changes, such as membrane ruffling and cytoplasmic blebbings or extrusions (*Figure 3*, *Figure 3—figure supplement 2*). These cytoplasmic changes in CPs recovered after U46619 washout (>30 min), suggesting that the transient U46619-mediated activation of CPs was not toxic. Nonetheless, to further investigate calcium signaling properties of CPs, we omitted the use of U46619 in our slice preparations. Furthermore, diameter measurements in brain slices were not performed because two-photon microscopy of stained stationary blood plasma may not reliably detect vessel borders.

Worthy of note is that in the absence of U46619 VPs also retained their calcium dynamics ex vivo with no evident change in both somata and processes (*Figure 3—figure supplement 3*), further emphasizing the viability of mural cells in our slice preparations and that mural cells differ in their calcium signaling properties.

## CP calcium events are evoked by vasomodulators

Next, we probed CPs for their responsiveness to different vasomodulators, such as G-protein-coupled receptor (GPCR) agonists Endothelin-1 (100 nM), UDP-Glucose (100 μM), and ATP (100 μM). Indeed, all agonists triggered a cytosolic calcium increase in CPs (*Figure 4A–C*), indicating that CPs express functional GPCRs and can respond to factors that are known to be released by endothelial cells or astrocytes (*Dehouck et al., 1997*; *Lazarowski and Harden, 2015*; *Harden et al., 2010*; *Koizumi et al., 2005*). Since the tested agonists activate GPCRs, mainly acting via phospholipase C and subsequent IP$_3$-mediated calcium release from internal stores (*Wynne et al., 2009*; *Lazarowski and Harden, 2015*; *Abbracchio et al., 2006*), we continued to examine the involvement of ion channels, which have been shown to affect cytosolic calcium levels in SMCs (*Hill-Eubanks et al., 2011*). Application of Nimodipine (100 μM), a blocker of L-type voltage-gated calcium channels (VGCC) involved in SMC contractions, led to a moderate reduction of basal calcium activity in CP somata (9.1 ± 2.6 vs. 7.6 ± 1.7, *Figure 4D*), but not in processes (39.1 ± 8.6 vs. 39.6 ± 7.8, *Figure 4D*). However, a transient receptor potential channel (TRPC) blocker (*Earley and Brayden, 2015*), SKF96365 (100 μM), reduced the calcium transient frequency more than threefold in both somata and processes (*Figure 4E*), suggesting that CPs require extracellular calcium to maintain their basal calcium fluctuations. Control drug vehicle (DMSO) experiments revealed no changes in CP calcium signal frequency in both somata and processes (*Figure 4—figure supplement 1*). Moreover, to rule out a signal run-down due to cell death when treated with SKF96365, cells were afterwards subjected to a brief stimulation with U46619, which triggered a robust calcium response in CPs (*Figure 4E*).

## Neuronal stimulation leads to a calcium signal drop in mural cells from capillaries to upstream arterioles

To investigate how acute changes in neuronal activity impact mural cell calcium dynamics, we used a chemogenetic approach (*Roth, 2016*), in

**Video 4.** Basal venular pericyte calcium activity in vivo. Recorded at 11.84 Hz and played at 45 fps, showing Merge, TxRed, and GCaMP6s channels. Scale bar 10 μm.

https://elifesciences.org/articles/70591/figures#video4

**Table 1.** Basal calcium signal properties of mural cells in awake and anesthetized animals.
Summary table of calcium signal frequency, amplitude, and duration of different mural cells from awake and anesthetized measurements in vivo. Values are represented as mean ± SD.

| | Frequency (signals/min) | | Amplitude (df/f) | | Duration (s) | |
|---|---|---|---|---|---|---|
| | Awake | Anesthetized | Awake | Anesthetized | Awake | Anesthetized |
| SMC | 9.4 ± 3.0 | 7.1 ± 1.6 | 0.3 ± 0.2 | 0.6 ± 0.3 | 3.1 ± 1.3 | 3.4 ± 0.9 |
| EP_Soma | 8.9 ± 3.2 | 8.3 ± 2.8 | 0.5 ± 0.4 | 0.6 ± 0.4 | 2.6 ± 0.9 | 3.0 ± 0.9 |
| EP_Process | 20.2 ± 7.8 | 21.6 ± 12.0 | 0.9 ± 0.5 | 1.3 ± 0.7 | 2.9 ± 0.7 | 3.4 ± 0.9 |
| CP_Soma | 6.9 ± 2.7 | 8.5 ± 2.3 | 0.3 ± 0.1 | 1.1 ± 1.3 | 2.4 ± 0.7 | 2.4 ± 0.6 |
| CP_Process | 39.6 ± 14.5 | 45.7 ± 19.3 | 0.7 ± 0.3 | 1.3 ± 0.4 | 2.6 ± 0.6 | 2.4 ± 0.4 |

which transduced neurons expressing hM3D(Gq)-DREADD (designer receptors exclusively activated by designer drugs) were activated with 30 µg/kg iv. clozapine (*Jendryka et al., 2019*). Clozapine was chosen over clozapine *N*-oxide (CNO) due to better bioavailability and the absence of side effects in the here applied concentration (*Gomez et al., 2017*; *Cho et al., 2020*). We injected 50 nl of AAV2-hSyn-hm3D(Gq)-mCherry into the somatosensory cortex with the aim to limit the transduced area as such, that along a continuous vascular branch the area of hM3D(Gq)-DREADD expressing neurons was primarily within the capillary bed (*Figure 5A and B*). This allowed us to investigate how neuronal activation alters mural cell calcium responses along a connected vascular branch from capillaries to arterioles and arteries.

Chemogenetic activation of neurons completely diminished calcium activity in CPs in both somata and processes (*Figure 5C*, *Video 6*). Possible non-specific effects caused by clozapine on CP calcium signals were ruled out because CPs outside the area of DREADD-expressing neurons retained their spontaneous calcium dynamics (*Figure 5—figure supplement 1*). A similar calcium drop in pericytes in response to neuronal activity was recently reported in the olfactory bulb upon odorant stimulation (*Rungta et al., 2018*). Interestingly, we observed that the drop in CP calcium signals was accompanied by a capillary diameter increase of 8% compared to baseline (*Figure 5D*). We further monitored calcium changes in upstream EPs and SMCs in the arterioles and pial artery connected to the investigated capillaries. These upstream segments were outside of the DREADD expressing area, however, although we could not detect mCherry signals we cannot exclude projections of neurons expressing low levels of hM3D(Gq) in these upstream segments. Nonetheless, in response to chemogenetic activation of neurons predominantly located in the vicinity of the capillary bed, also EPs exhibited a clear drop in calcium signals in both somata and processes (*Figure 5E*). And the diameter of EP-associated arterioles increased by 22% compared to baseline (*Figure 5F*). Furthermore, in the upstream connected pial artery, SMCs also exhibited a calcium signal drop (*Figure 5G*) and arterial diameter increased by about 19% (*Figure 5H*). Importantly, to rule out that the calcium signal drop is not caused by toxicity associated with clozapine induced DREADD activation, we monitored the same cells one day after the clozapine treatment and found a complete recovery of calcium activity in all of the observed mural cells (*Figure 5—figure supplement 2*). Thus, chemogenetic neuronal activation led to a pronounced calcium drop in all mural cells along the same vascular tree and to an increase in vessel diameters, from the capillary to the surface pial artery.

## Elevations in extracellular potassium lead to a calcium signal drop in CPs

When stimulated, neurons release potassium into the extracellular space (*Somjen, 1979*; *Heinemann and Lux, 1977*). This potassium is sensed by SMCs, causing the suppression of calcium oscillations (*Filosa et al., 2006*). To test whether calcium signals also drop in CPs to elevations in extracellular potassium, we temporarily raised the extracellular potassium concentration in acute brain slices by 10 mM (in the form of KCl). Indeed, CPs revealed a pronounced drop in calcium activity in both somata and processes in response to potassium stimulation (*Figure 6A*). To exclude any secondary stimulation effects, neuronal firing activity was blocked with 1 µM tetrodotoxin (TTX) (*Figure 6—figure*

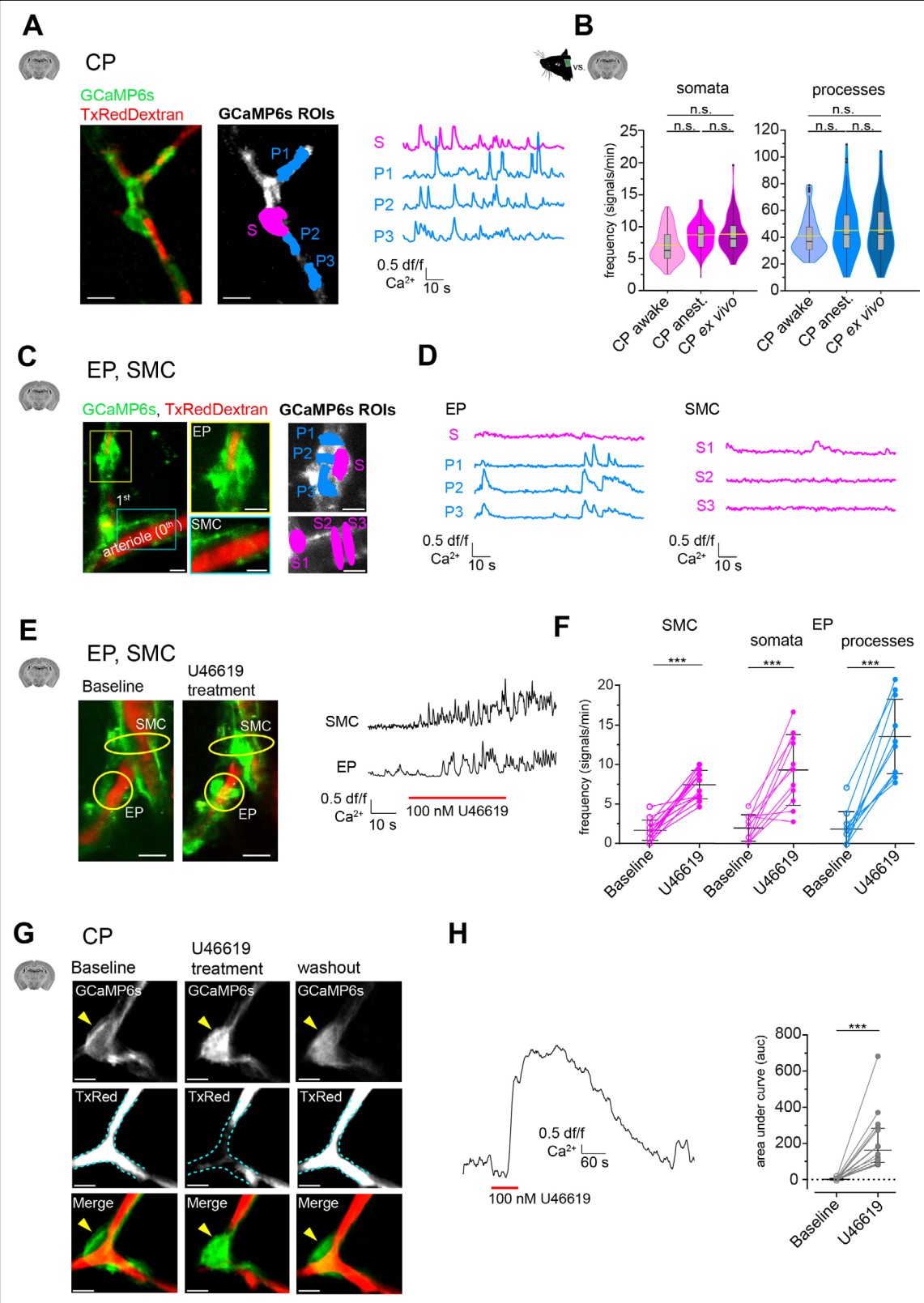

**Figure 3.** Mural cell calcium dynamics in acute cortical brain slices. (**A**) Representative image of a CP ex vivo. In the GCaMP6s channel, ROIs for soma (S, in magenta) and processes (P1–3, in blue) are shown. On the right are the respective normalized traces of calcium signals. (**B**) Violin/box plots depicting the quantified calcium signal frequency of CPs ex vivo compared to the previously (**Figure 2**) determined calcium signal frequency in vivo for somata and processes. The dashed yellow lines indicate the mean values. Statistics were calculated using linear mixed-effects models and Tukey post hoc tests,

*Figure 3 continued*

$CP_{awake}$: N = 3, n = 96; $CP_{anest}$: N = 25, n = 93; $CP_{ex\ vivo}$: N = 52, n = 106, n.s. indicates not significant. Scale bars: 10 µm. (**C**) SMCs and EP ex vivo can be localized similarly to those in vivo by following the vessel branches of a pial artery lying on the surface of the brain slice. Z-stack (20 µm) image of an arteriole and its adjacent 1st-order branch, where SMCs and an EP is located. The yellow box shows a magnified image of the EP and the cyan box shows a magnified image of SMCs. In the GCaMP6s channel, ROIs for soma (S, in magenta) and processes (P1–3, in blue) are shown. (**D**) Corresponding normalized traces of calcium signals for the EP and SMCs in (**C**) are shown. (**E**) Z-stack (20 µm) images of an arteriole and its adjacent 1st-order branch harboring SMCs and an EP (indicated by the yellow circles), before and after U46619 treatment. Next to the images are the corresponding normalized traces of calcium signals for the indicated SMC and EP. (**F**) Quantifications of the calcium signal frequency in SMC and EP somata and processes, comparing baseline to the U46619 treatment. Data represents individual cells, median and interquartile ranges. Statistics were calculated using two-tailed paired t-tests, SMC: N = 4, n = 14; EP: N = 4, n = 11, ***p<0.001. Scale bars: 10 µm. (**G**) Time-averaged (10 s) images of a CP before, during U46619 (100 nM) treatment and recovery (30 min washout of U46619). U46619 causes a massive calcium response, which is accompanied by cytoplasmic blebbing and membrane ruffling (yellow arrowhead). The dashed cyan lines in the TxRed images outline the vessel at baseline level to highlight vessel changes during treatment and recovery. (**H**) A normalized trace of the calcium response of a CP to U46619 (100 nM) is shown. On the right is the quantification of the area under the curve (auc) comparing baseline to U46619 treatment. Data represents individual cells, median and interquartile ranges. Statistics were calculated using a two-tailed Wilcoxon matched-pairs signed rank test. N = 5, n = 16, ***p<0.001. Scale bars: 5 µm. The red line below the calcium trace indicates the time of drug addition. See also *Figure 3—figure supplements 1–3*. Data: *Figure 3—source data 1*.

The online version of this article includes the following source data and figure supplement(s) for figure 3:

**Source data 1.** Source data for *Figure 3*.

**Figure supplement 1.** CP calcium signal properties in vivo compared to ex vivo.

**Figure supplement 1—source data 1.** Source data for *Figure 3—figure supplement 1*.

**Figure supplement 2.** U46619 induced cytoplasmic changes in CPs.

**Figure supplement 3.** VP calcium signals in vivo compared to ex vivo.

**Figure supplement 3—source data 1.** Source data for *Figure 3—figure supplement 3*.

*supplement 1*). TTX alone did not influence the calcium dynamics of CPs in both somata and processes (*Figure 6—figure supplement 2*).

In SMCs and ECs, Kir2 channels are the main drivers for potassium sensing (*Longden and Nelson, 2015*; *Longden et al., 2017*). Single-cell transcriptomic analysis revealed a high abundance of Kir2.2 (*Kcnj12*) transcripts in pericytes (*Vanlandewijck et al., 2018*). Hence, it could be that a similar mode of potassium sensing occurs also in pericytes. We antagonized Kir2 channels with $BaCl_2$ (100 µM), and interestingly, we observed a less pronounced decrease in CP calcium frequency in response to potassium stimulation in both somata (7.7 ± 2.5 vs. 4.6 ± 2.7) and processes (55.1 ± 27.9 vs. 43.2 ± 33.2, *Figure 6B*).

Furthermore, an ATP-sensitive potassium channel (*Nichols, 2006*) composed of Kir6.1 (*Kcnj8*) and its associated sulfonylurea receptor 2 (SUR2, *Abcc9*) has been suggested as a marker for brain pericytes due to its high abundance in CPs (*Bondjers et al., 2006*; *Vanlandewijck et al., 2018*) and was recently shown to be involved in CP hyperpolarization (*Sancho et al., 2021*). Application of PNU-37883A (100 µM), a Kir6.1 specific antagonist (*Teramoto, 2006*), together with the Kir2 channel blocker $BaCl_2$ abolished the decrease in CP calcium signals in response to potassium stimulation, in both somata (9.1 ± 3.2 vs. 9.0 ± 4.0) and processes (46.9 ± 31.1 vs. 55.0 ± 36.4, *Figure 6C*).

Next, we investigated the possible involvement of $K_{ATP}$ channels in the regulation of calcium signaling in CPs. Lowering the cellular [ATP]/[ADP] ratio by blocking mitochondrial activity with sodium azide ($NaN_3$, 5 mM) in ex vivo brain slices should result in $K_{ATP}$ channel activation (*Dart and Standen, 1995*; *Tinker et al., 2018*; *Reiner et al., 1990*; *Lerchundi et al., 2019*). Indeed, CP calcium signals decreased strongly in response to $NaN_3$ in both somata and processes (*Figure 6D*). To test whether reducing mitochondrial respiration impacts CP calcium dynamics also in vivo, we exposed anesthetized animals to an acute hypoxia insult, by replacing oxygen in the anesthesia gas mixture transiently with pure nitrogen. Indeed, transient hypoxia led to a complete cessation of CP calcium activity in both somata and processes

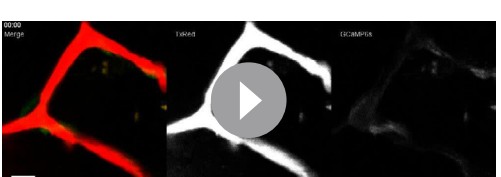

**Video 5.** Ex vivo application of U46619 (100 nM) leads to a strong calcium response in CPs with cytoplasmic extrusions. Recorded at 0.75 Hz and played at 20 fps, showing Merge, TxRed, and GCaMP6s channels. Scale bar 10 µm.
https://elifesciences.org/articles/70591/figures#video5

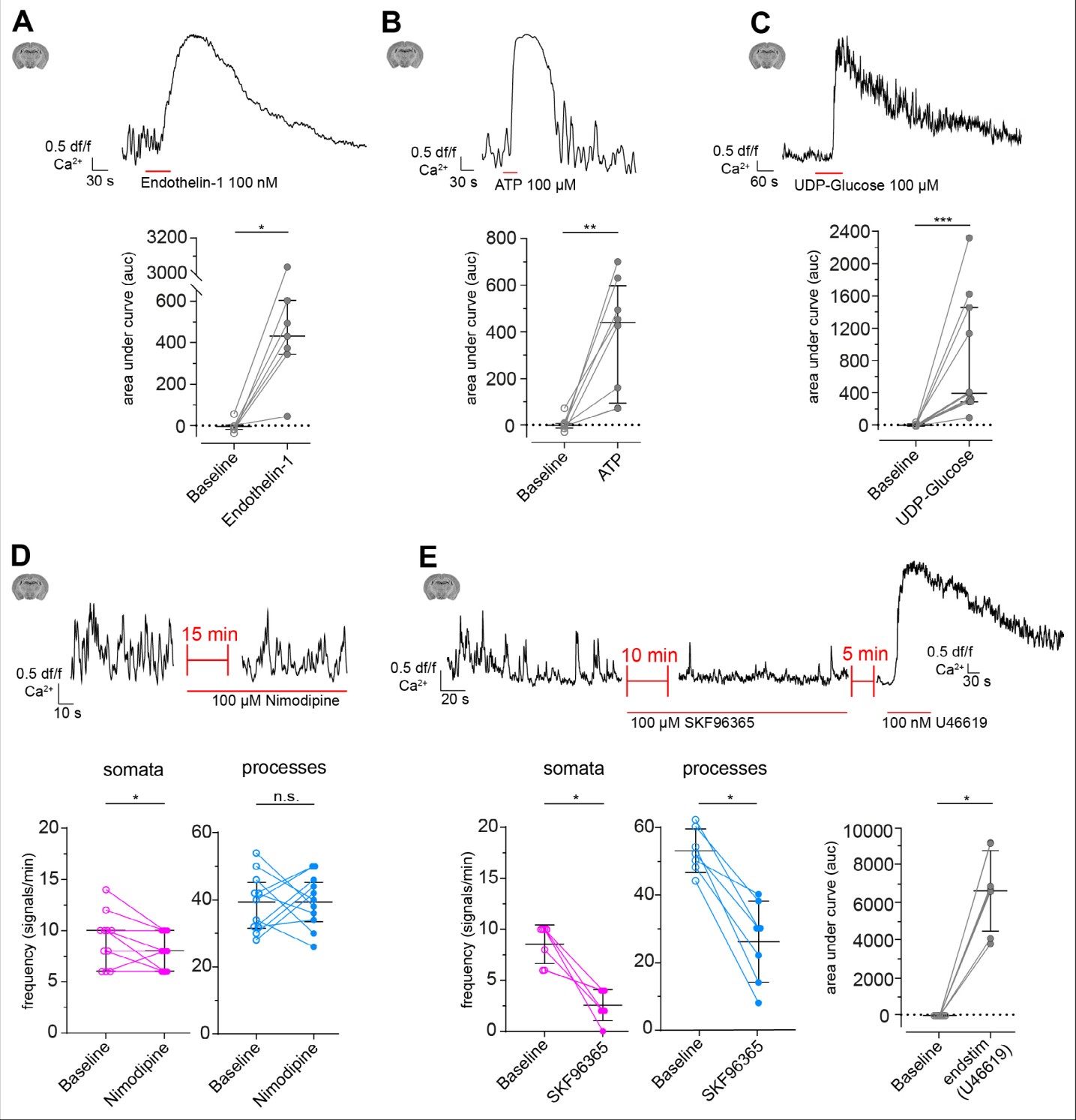

**Figure 4.** Modulation of CP calcium signals ex vivo. (**A–C**) CP calcium responses to the application of vasomodulators: (**A**) Endothelin-1 (100 nM), (**B**) ATP (100 µM), and (**C**) UDP-Glucose (100 µM). On top are representative normalized calcium signal traces and below are quantifications of the area under the curve (auc), comparing baseline to the respective treatment. Data represents individual cells, median and interquartile ranges. Statistics were calculated using Wilcoxon matched-pairs signed rank tests. Endothelin-1: N = 3, n = 7, *p=0.02; ATP: N = 4, n = 8, **p=0.008; UDP-Glucose: N = 4, n = 11, ***p<0.001. (**D**) CP calcium response to application of the L-type voltage-gated calcium channel (L-type VGCC) blocker Nimodipine (100 µM). On top is a representative normalized calcium signal trace and below is the quantification of the signal frequency in somata and processes, comparing baseline to the Nimodipine treatment. Nimodipine was infused 15 min prior to data collection. Data represents individual cells, median and interquartile ranges. Statistics were calculated using two-tailed paired t-tests, N = 4, n = 11. S: t(10) = 2.39, *p=0.04; P: t(10) = 0.1495, p=0.88. (**E**) CP calcium response

*Figure 4 continued on next page*

*Figure 4 continued*

to application of the TRPC channel blocker SKF96365 (100 μM) and endstimulus application (U46619, 100 nM). On top is a representative normalized calcium signal trace and below is the quantification of the signal frequency in somata and processes, comparing baseline to the SKF96365 treatment and the quantification of the area under the curve (auc), comparing baseline to endstimulus treatment. Data represents individual cells, median and interquartile ranges. Statistics were calculated using a Wilcoxon matched-pairs signed rank test, N = 3, n = 7. S: *p=0.02; P: *p=0.02, endstim: *p=0.02. The red lines below the traces indicate the addition time of the respective drug. n.s. indicates not significant, *p<0.05, **p<0.01, ***p<0.001. See also *Figure 4—figure supplement 1*, 2 and 3. Data: *Figure 4—source data 1*.

The online version of this article includes the following source data and figure supplement(s) for figure 4:

**Source data 1.** Source data for *Figure 4*.

**Figure supplement 1.** CP calcium response to application of DMSO (0.2%).

**Figure supplement 1—source data 1.** Source data for *Figure 4—figure supplement 1*.

(*Figure 6E*). When oxygen supply was re-established, calcium activity in CPs was resumed. This hypoxia-induced drop in calcium activity was specific to mural cells, since nearby GCaMP6s-labeled astrocytes reacted with a strong increase in calcium levels upon hypoxia (*Figure 6E*). Additionally, we assessed whether capillary diameters changed during the acute hypoxic challenge. Indeed, during hypoxia capillary diameters dilated on average by 11% compared to baseline (*Figure 6—figure supplement 3*). When re-establishing the oxygen supply, capillary diameters returned to baseline values. Overall, these capillary diameter changes coincided with the CP calcium signal changes during and after hypoxia (*Figure 6E*, *Figure 6—figure supplement 3*).

## Discussion

The present study identifies distinct calcium signatures of different mural cells in the somatosensory cortex of healthy mice. To differentiate between these phenotypically diverse mural cells, we employed a previously coined classification based on microvascular localization: SMC, EP, and CP (*Grant et al., 2019*; *Berthiaume et al., 2021*). Although phenotypic diversity of mural cells was already outlined in the earliest descriptions of pericytes almost a century ago (*Zimmermann, 1923*), there is still no consensus on mural cell classifications in the brain (*Attwell et al., 2016*). Especially mural cells located at the terminal end of arterioles (1st–4th branch order) are not clearly defined and were variously named in recent literature as precapillary/terminal SMCs (*Hill et al., 2015*), contractile mural cells (*Hariharan et al., 2020*), contractile pericytes (*Gonzales et al., 2020*), or EPs (*Berthiaume et al., 2021*). Whether these cells are a subtype of SMCs or pericytes remains to be elucidated.

CPs have been reported to have frequent calcium transients in somata and in microdomains along processes (*Hill et al., 2015*; *Rungta et al., 2018*). We have extended these observations and found that all mural cells along the arteriovenous axis exhibit basal calcium fluctuations (*Figure 2A–D*, *Video 1*, *Video 2*, *Video 3*, *Video 4*). During spontaneous vasomotor activity, SMCs exhibited calcium oscillations that are inversely correlated to vessel diameter changes (*Figure 2E*). These rhythmic pulsations were shown to be based on entrainment by ultra-slow fluctuations (0.1 Hz) in γ-band neuronal activity (*Mateo et al., 2017*). We observed the same behavior of calcium oscillations in EPs on penetrating arterioles (1st–4th branch order, *Figure 2F*). This is in line with the expression of α-smooth muscle actin in mural cells of up to 4th branch order vessels as seen by immunohistochemistry staining and the use of transgenic animals (*Hartmann et al., 2015a*; *Hill et al., 2015*). In recent computational and experimental works, these vascular segments have also been shown to be the location of blood flow regulation to supply a particular downstream region (*Sweeney et al., 2018*; *Grubb et al., 2019*; *Gonzales et al., 2020*; *Video 7*). Thus, it is likely that conclusions in implicating pericytes in the control of blood flow in several studies were based on experiments focusing on EPs (*Peppiatt et al., 2006*; *Hall et al., 2014*). At the capillary level, which we defined as vessels with an average diameter of 4 μm and a branch order of >4, CPs exhibit asynchronous calcium signals between somata and processes (*Figure 2C*). In agreement with previous studies, we did not find an immediate correlation between vessel diameter and calcium transients in CPs in vivo under basal conditions (*Figure 2G*; *Hill et al., 2015*; *Rungta et al., 2018*). Nonetheless, we have to note that two-photon imaging is not suitable for the detection of minute and fast changes in vessel diameter in the expected range of 1–3% for capillaries (*Grutzendler and Nedergaard, 2019*).

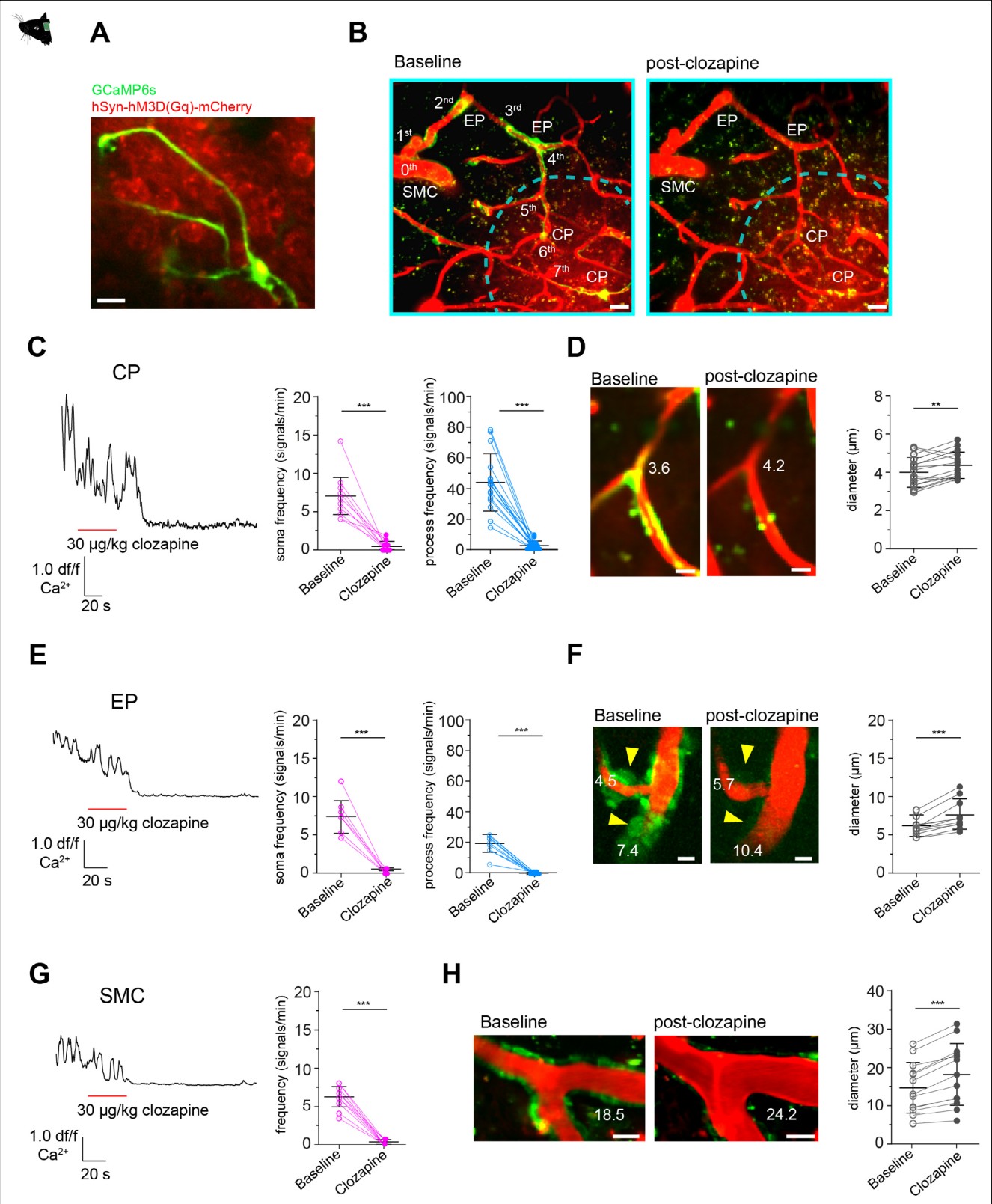

**Figure 5.** Calcium signal drop in all mural cells along a vascular branch in response to neuronal activation. (**A**) Z-stack (10 μm) image of a CP inside the area of neurons, expressing hM3D(Gq)-mCherry. (**B**) Z-stack (75 μm) of a vascular tree with connected capillaries reaching into an area of neurons expressing hM3D(Gq)-mCherry (indicated by the dashed cyan lines), before and during clozapine treatment. Scale bar = 10 μm. (**C**) Normalized calcium signal trace of a CP to chemogenetic activation of hM3D(Gq) transduced neurons in vivo and the quantification of the calcium signal frequency,

*Figure 5 continued on next page*

*Figure 5 continued*

comparing baseline to stimulation. Data represents individual cells, median, and interquartile ranges. Statistics were calculated using a Wilcoxon matched-pairs signed rank test for S (***p<0.001) and a two-tailed paired t-test for P (t(15) = 8.815, ***p<0.001), N = 4, n = 16. (**D**) Images of a CP before and during clozapine treatment. The numbers indicate the vascular diameter in μm. On the right vascular diameters measured at the site of the soma, before (4.1 ± 0.8 μm) and during treatment (4.5 ± 0.7 μm) are shown. Data represents individual cells, mean and standard deviation. Statistics were calculated using a two-tailed paired t-test, t(15) = 3.245, **p=0.005, N = 4, n = 16. Scale bar = 5 μm. (**E**) Normalized calcium signal trace of an EP (located on a vascular branch connected to capillaries in an area of hM3D(Gq) transduced neurons) to chemogenetic activation of neurons in vivo. On the left quantification of the calcium signal frequency, comparing baseline to stimulation is shown. Data represents individual cells, median and interquartile ranges. Statistics were calculated using a two-tailed paired t-test for S (t(10) = 10.71, ***p<0.001) and a Wilcoxon matched-pairs signed rank test for P (***p<0.001), N = 3, n = 11. (**F**) Images of two EPs (somata are indicated by yellow arrowheads) before and during clozapine treatment are shown. The numbers indicate the vascular diameter in μm. On the right vascular diameters measured at the site of the soma, before (5.9 ± 1.1 μm) and during treatment (7.5 ± 1.9 μm) are shown. Data represents individual cells, mean and standard deviation. Statistics were calculated using a two-tailed paired t-test, t(10) = 5.375, ***p<0.001, N = 3, n = 11. Scale bar = 5 μm. (**G**) Normalized calcium signal trace of a SMC (located on a vascular branch connected to capillaries in an area of hM3D(Gq) transduced neurons) to chemogenetic activation of neurons in vivo. On the left quantification of the calcium signal frequency, comparing baseline to stimulation is shown. Data represents individual cells, median and interquartile ranges. Statistics were calculated using a two-tailed paired t-test, t(12) = 16.55, ***p<0.001, N = 3, n = 13. (**H**) Images of SMCs before and during clozapine treatment are shown. The numbers indicate the vascular diameter in μm. On the right vascular diameter before (14.7 ± 6.6 μm) and during treatment (18.2 ± 8.1 μm) are shown. Data represents individual cells, mean and standard deviation. Statistics were calculated using a two-tailed paired t-test, t(12) = 6.985, ***p<0.001, N = 3, n = 13. Scale bar = 20 μm. See also *Figure 5—figure supplements 1 and 2*. Data: *Figure 5—source data 1*.

The online version of this article includes the following source data and figure supplement(s) for figure 5:

**Source data 1.** Source data for *Figure 5*.

**Figure supplement 1.** CP calcium frequency is not affected by clozapine.

**Figure supplement 1—source data 1.** Source data for *Figure 5—figure supplement 1*.

**Figure supplement 2.** Recovery of mural cell calcium activity post-clozapine treatment.

Interestingly, CP calcium signal frequency persisted in ex vivo slice experiments with no substantial difference from the signal frequency in vivo (*Figure 3A and B*). This could hint to a blood flow independent signal, which is rather influenced by the content of solutes and oxygen, available in excess to the cells in the slice preparation. In contrast, SMCs and EPs showed a pronounced reduction in their calcium signal frequency ex vivo (*Figure 3C and D*). Due to the lack of intraluminal pressure in acute brain slices, arteries and arterioles lose basal tone, resulting in vascular collapse (*Mishra et al., 2014*). This likely led to cessation of basal calcium oscillations in SMCs and EPs ex vivo (*Figure 3C and D*). Addition of a widely used preconstricting factor (U46619, 100 nM) to the brain slice (*Brown et al., 2002*; *Filosa et al., 2006*) restored calcium activity in both SMCs and EPs (*Figure 3E and F*). However, we found that addition of U46619 caused CPs to form membranous blebs and cellular extrusions as well as provoking a large cytosolic calcium increase (*Figure 3G and H*, *Video 5*). This aberrant calcium response was accompanied by vessel constriction, which were also reported earlier (*Fernández-Klett et al., 2010*). It is likely that CPs are able to exert force on their underlying vessel in pathological conditions via cytoskeletal rearrangements. A comparable scenario was suggested in Aβ-induced constriction of capillaries in Alzheimer's disease (*Nortley et al., 2019*). Moreover, high-power optogenetic stimulation of CPs produced a similar effect (*Hartmann et al., 2021*).

Furthermore, calcium is involved in a plethora of cellular processes, ranging from cellular homeostasis to force generation (*Berridge et al., 2003*). This may explain the spread of calcium signals observed in CPs at basal conditions. The oscillatory nature of the CP calcium dynamic could be a way of reducing the threshold for activation of

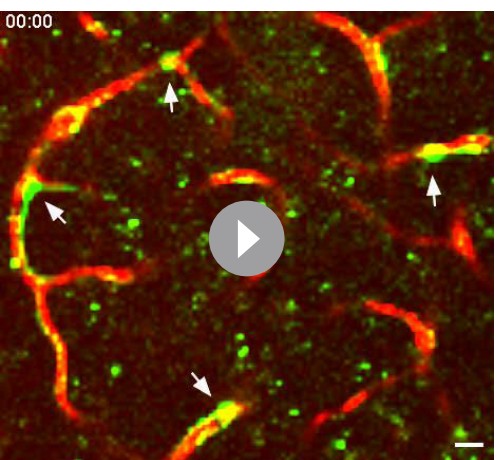

**Video 6.** Chemogenetic activation of hM3D(Gq) DREADD transduced neurons leads to a calcium signal drop in CPs. Arrows point to CPs. Recorded at 11.84 Hz and played at 45 fps. Scale bar 10 μm.

https://elifesciences.org/articles/70591/figures#video6

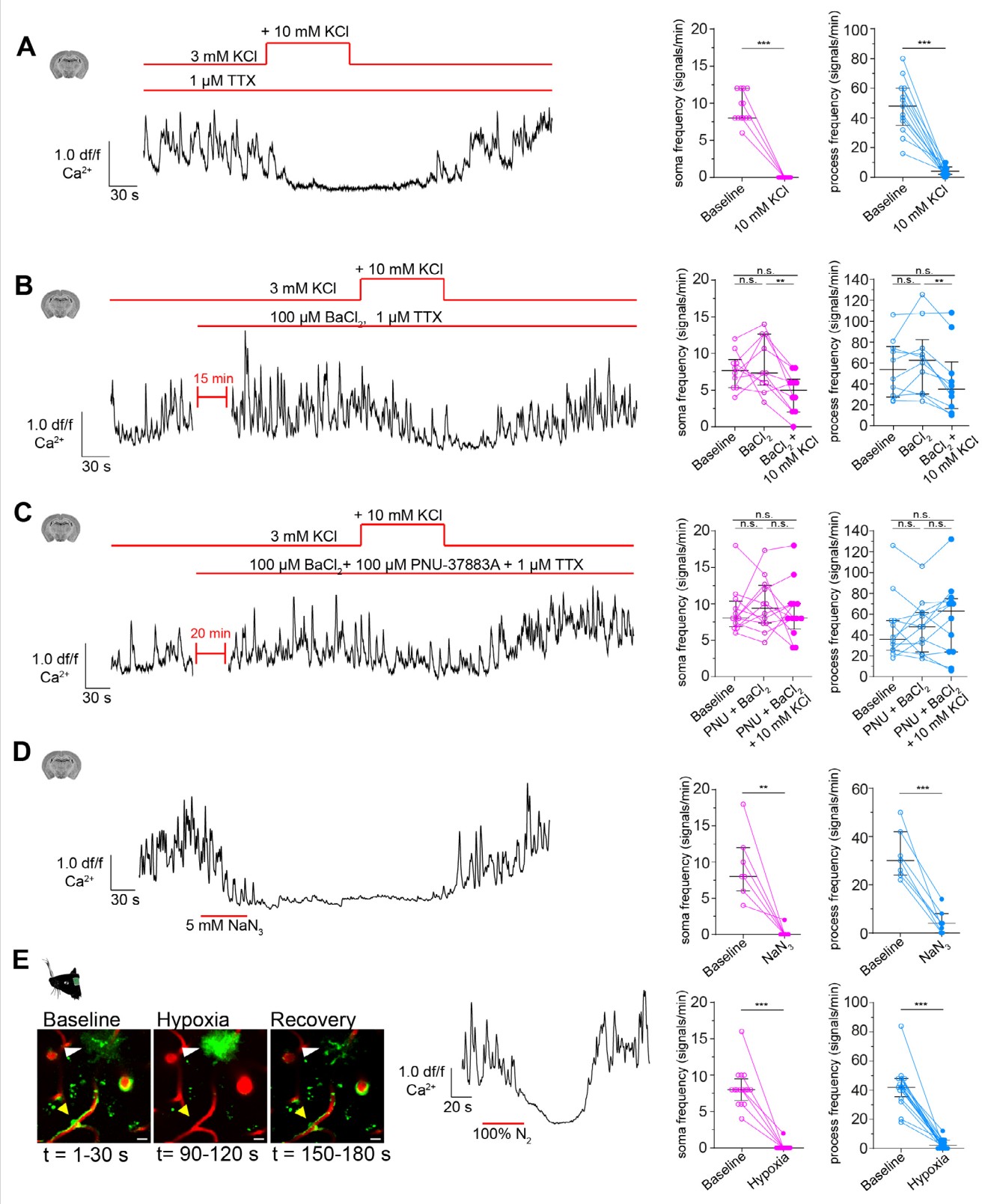

**Figure 6.** CPs react to increased extracellular potassium. (**A**) Calcium response of CPs to a 10 mM rise of extracellular potassium in the presence of TTX (1 µM) ex vivo. A representative normalized calcium signal trace (left) and the quantification of the calcium signal frequency, comparing baseline to treatment (right) are shown. Data represents individual cells, median and interquartile ranges. Statistics were calculated using a Wilcoxon matched-pairs signed rank test (for S) and a two-tailed paired t-test (for P), N = 4, n = 13. S: ***p<0.001; P: t(12) = 8.516, ***p<0.001. (**B**) Calcium response of CPs

*Figure 6 continued on next page*

*Figure 6 continued*

to a 10 mM rise of extracellular potassium in the presence of TTX (1 µM) and Kir2 channel blocker $BaCl_2$ (100 µM) ex vivo. A representative normalized calcium signal trace is shown on the left and on the right, the calcium signal frequency is quantified, comparing baseline to pre-treatment (TTX + $BaCl_2$) and additional potassium treatment. Data represents individual cells, median and interquartile ranges. Statistics were calculated using a one-way ANOVA, N = 4, n = 10. S: $F_{(1.796, 16.17)}$ = 5.728, p=0.02, P(baseline vs. $BaCl_2$) = 0.82, p(baseline vs. $BaCl_2$+ KCl) = 0.10, p($BaCl_2$ vs. $BaCl_2$+ KCl) = 0.009; P: $F_{(1.542, 13.88)}$ = 7.137, p=0.01, p(baseline vs. $BaCl_2$) = 0.31, p(baseline vs. $BaCl_2$+ KCl) = 0.22, P($BaCl_2$ vs. $BaCl_2$+ KCl) = 0.002. (**C**) Calcium response of CPs to a 10 mM rise of extracellular potassium in the presence of TTX (1 µM), Kir2 channel blocker BaCl2 (100 µM), and KATP channel blocker PNU-37883A (100 µM) ex vivo. A representative normalized calcium signal trace is shown on the left. On the right, the calcium signal frequency is quantified, comparing baseline to pre-treatment (TTX + $BaCl_2$+ PNU-37883A) and additional potassium treatment. Data represents individual cells, median and interquartile ranges. Statistics were calculated using a one-way ANOVA, N = 4, n = 12. S: $F_{(1.395, 15.35)}$ = 0.2248, p=0.72, P(baseline vs. PNU + $BaCl_2$) = 0.76, p(baseline vs. PNU + $BaCl_2$+ KCl) = 0.99, p(PNU + $BaCl_2$ vs. PNU + $BaCl_2$+ KCl) = 0.77; P: $F_{(1.683, 18.51)}$ = 0.9687, p=0.38, p(baseline vs. PNU + $BaCl_2$) = 0.96, P(baseline vs. PNU + $BaCl_2$+ KCl) = 0.54, P(PNU + $BaCl_2$ vs. PNU + $BaCl_2$+ KCl) = 0.44. (**D**) Calcium response of CPs to $NaN_3$ (5 mM) ex vivo. A representative normalized calcium signal trace is shown on the left and the quantification of the calcium signal frequency, comparing baseline to $NaN_3$ treatment is shown on the right. Data represents individual cells, median and interquartile ranges. Statistics were calculated using two-tailed paired t-tests, N = 3, n = 7. S: $t_{(6)}$ = 4.824, **p=0.003; P: $t_{(6)}$ = 7.621, ***p<0.001. (**E**) Calcium response of CPs to an acute hypoxia insult in vivo. On the left, two-photon images show the time course and GCaMP6s fluorescence of the hypoxic insult (yellow arrowhead: capillary pericyte; white arrowhead: astrocyte). Center, a representative normalized calcium signal trace is shown. The calcium signal frequency is quantified on the right. Baseline is compared to hypoxic intervention. Data represents individual cells, median and interquartile ranges. Statistics were calculated using Wilcoxon matched-pairs signed rank tests, N = 4, n = 16. S: ***p<0.001; P: ***p<0.001. The red lines indicate the addition time of the respective drug. See also ***Figure 6—figure supplements 1–3***. Data: ***Figure 6—source data 1***.

The online version of this article includes the following source data and figure supplement(s) for figure 6:

**Source data 1.** Source data for *Figure 6*.

**Figure supplement 1.** Electrophysiology of cortical L2/3 pyramidal neurons.

**Figure supplement 1—source data 1.** Source data for *Figure 6—figure supplement 1*.

**Figure supplement 2.** CP calcium response to application of TTX (1 µM).

**Figure supplement 2—source data 1.** Source data for *Figure 6—figure supplement 2*.

**Figure supplement 3.** Capillary diameter changes to hypoxic insults.

**Figure supplement 3—source data 1.** Source data for *Figure 6—figure supplement 3*.

---

calcium-related pathways, such as regulation of gene expression (***Dolmetsch et al., 1998***). Regarding the importance of calcium in the contractile machinery of SMCs (***Goulopoulou and Webb, 2014***), a relationship between intracellular calcium in CPs and vasoactivity is possible: We observed that elevations of calcium in CPs were associated with constriction (***Figure 3G***, ***Figure 3—figure supplement 2***), while drops in calcium were associated with dilations (***Figure 5D***, ***Figure 6—figure supplement 3***). This suggests that CPs may be able to exert calcium-related forces on capillaries with possible impact on blood flow regulation. Just recently it could be shown that optogenetic stimulation of CPs could induce capillary constriction. These constrictions could be prevented by fasudil, an inhibitor of actomyosin–cytoskeleton regulatory rho-kinases (***Hartmann et al., 2021***). Since expression of smooth muscle actin in CPs seems absent or very low (***Grant et al., 2019***; ***Alarcon-Martinez et al., 2018***), the type of contractile machinery in CPs remains to be elucidated. Further knowledge of the pericyte calcium-signaling toolkit (***Hariharan et al., 2020***) may help to understand the consequences of altered calcium signaling in pericytes in physiological and pathophysiological conditions.

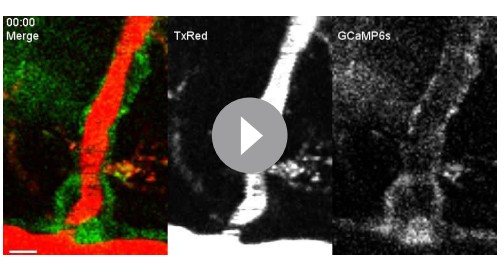

**Video 7.** Mural cells at a precapillary branch. Recorded at 11.84 Hz and played at 45 fps, showing Merge, TxRed, and GCaMP6s channels. Scale bar 10 µm.
https://elifesciences.org/articles/70591/figures#video7

Potassium is released in high amounts during the repolarization phase after action potential firing (***Paulson and Newman, 1987***) and can act as a potent vasomodulator (***McCarron and Halpern, 1990***). Potassium sensing by SMCs and capillary ECs via Kir2 channels has been previously described as a mechanism to increase local cerebral blood flow (***Longden and Nelson, 2015***; ***Longden et al., 2017***; ***Filosa et al., 2006***; ***Haddy et al., 2006***). Furthermore, a modeling study showcases the capillary EC Kir2 channel as a sensor of neuronal activity and highlights its impact on potassium-mediated neurovascular

communication (*Moshkforoush et al., 2020*). However, so far the role of CPs in this potassium-mediated vascular response remains elusive. Previous studies revealed a strong correlation between CP loss and dysregulated neurovascular coupling (*Kisler et al., 2017*; *Nikolakopoulou et al., 2019*). A recent study, using a pressurized retina preparation, intimates that junctional pericytes act as control elements in the potassium-mediated functional hyperemia by directing blood flow via branch-specific dilation to a site where a stimulus was evoked (*Gonzales et al., 2020*). Moreover, optogenetic stimulation of CPs was recently shown to induce capillary constriction (*Hartmann et al., 2021*), suggesting that CPs could be involved in neurovascular coupling.

In our study, we report that specific excitatory chemogenetic stimulation of neurons in vivo decreases calcium signals in all mural cells along a vascular branch and this was associated with an increase in vascular diameters including capillaries (*Figure 5*, *Video 6*). Comparable calcium signal reductions were found by odorant-stimulated neurons (*Rungta et al., 2018*) or during spreading depolarization (*Khennouf et al., 2018*). However, the threshold of neuronal excitation to cause a calcium response in CPs still needs to be further determined, since another study could not observe a calcium signal change in CPs in response to sensory whisker stimulation (*Hill et al., 2015*). We also demonstrate that CP calcium signals decrease in response to an acute rise of extracellular potassium concentration in acute brain slices (*Figure 6A*). Pharmacological inhibition of Kir2 and $K_{ATP}$ channels inhibited the potassium evoked calcium drop, implying a role of these channels in silencing calcium activity in CPs (*Figure 6B and C*). Moreover, both in vivo hypoxia and blockade of mitochondrial respiration ex vivo, presumably leading to a reduced intracellular [ATP]/[ADP] ratio and likely elevated potassium levels due to neuronal depolarization, also resulted in cessation of CP calcium signals (*Figure 6D and E*), possibly as a result of Kir2 and $K_{ATP}$ channel activation. High abundance of Kir2 and $K_{ATP}$ channel transcripts were reported in CPs (*Vanlandewijck et al., 2018*; *Bondjers et al., 2006*). Interestingly,

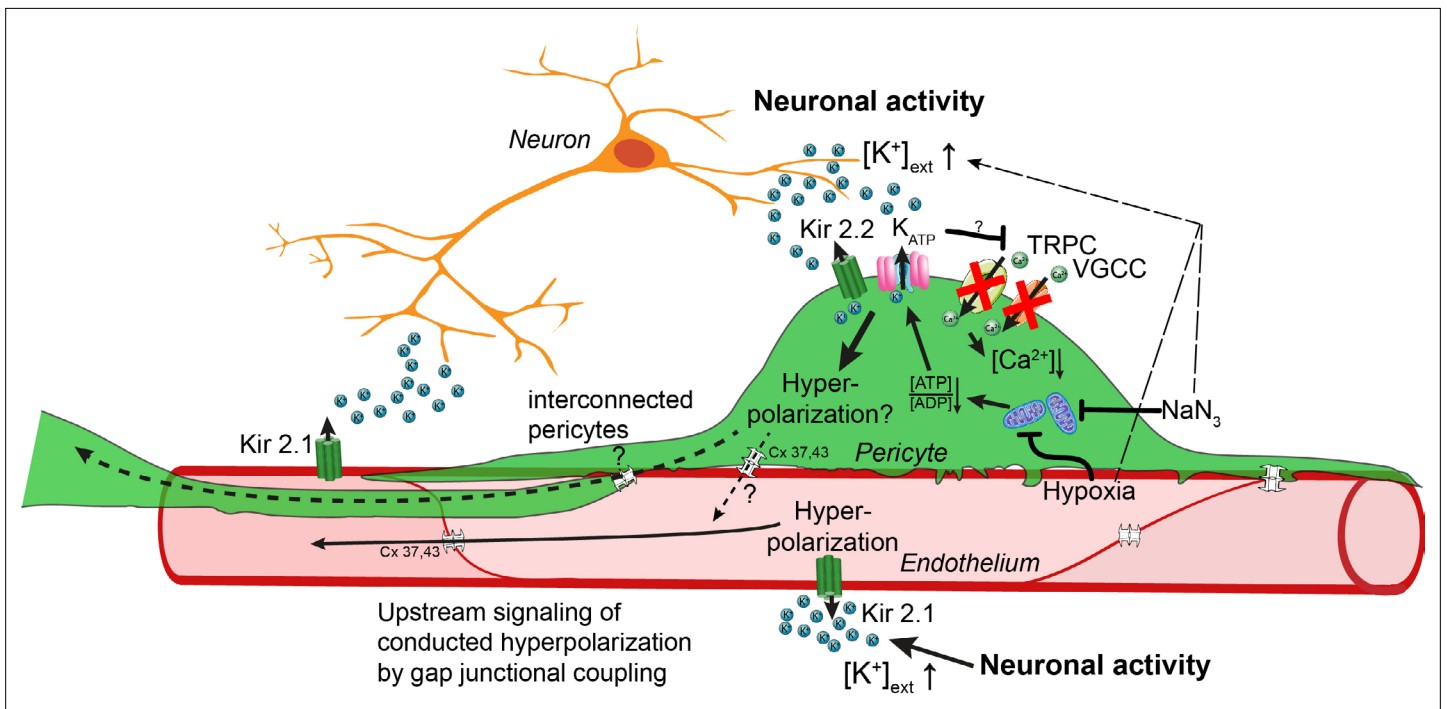

**Figure 7.** Working model of potassium-induced calcium changes in capillary pericytes. Neuronal activity leads to the release of potassium into the extracellular space. This rise in potassium [K⁺] activates Kir2.1 channels on capillary endothelial cells (EC) to induce a retrograde propagating hyperpolarization via gap-junctional coupling (*Longden et al., 2017*), which leads to upstream dilation of arteries. Elevated neuronal activity and a rise in extracellular potassium decrease calcium signaling in pericytes. Kir2.2 and $K_{ATP}$ channels expressed on capillary pericytes (CPs) may induce a hyperpolarization in CPs as recently shown (*Sancho et al., 2021*). This hyperpolarization would inactivate TRPC and VGCC channels in CPs leading to a drop in calcium signals. Hypoxia or inhibition of respiratory metabolism reduce the [ATP]/[ADP] ratio thereby activating $K_{ATP}$ and Kir2.2 channels to induce a hyperpolarization. By gap-junction coupling (Cx 37, 43) between ECs and CPs, the hyperpolarization of the EC–CP capillary unit could be transmitted retrogradely to induce upstream vascular responses.

a recent study could show that K_ATP channel activation leads to a hyperpolarization in CPs in retinal preparations (*Sancho et al., 2021*).

However, Kir2 and K_ATP channels are also expressed in capillary ECs, which mediate a hyperpolarization in response to increases in extracellular potassium (*Longden et al., 2017*). Given that there is evidence of EC and CP gap-junctional coupling in the retinal vasculature (*Wu et al., 2006*; *Kovacs-Oller et al., 2020*; *Ivanova et al., 2017*; *Ivanova et al., 2019*) and the lack of cell-type specificity of pharmacological manipulations, we cannot exclude that the observed potassium evoked calcium drop in CPs could be secondary to changes in capillary ECs. However, still very little is known of how capillary ECs interact and modulate CP functions or vice versa. Gap-junction coupling of capillary ECs and CPs would allow for electrical interconnection between these cells, forming a vascular relay (*Ivanova et al., 2019*). Activation of Kir2 and K_ATP channels likely induces cellular hyperpolarization in this EC–CP capillary unit, which could be transmitted between and within CPs and ECs through gap-junctional coupling (*Figure 7*). Potassium sensing by CPs and ECs could potentiate the propagation of hyperpolarization from a site of elevated neuronal activity to upstream feeding arterioles.

Moreover, CPs are optimally positioned in the capillary bed (like cellular antennas of the EC–CP unit) to sense the microenvironment (*Pfeiffer et al., 2021*) and to amplify the hyperpolarization-mediated vascular response. Future studies using concurrent calcium imaging in ECs and CPs as well as cell-specific knock-out models are needed to gain more insights into the individual and the intercellular contribution of capillary ECs and CPs in potassium sensing and neurovascular coupling.

# Materials and methods

**Key resources table**

| Reagent type (species) or resource | Designation | Source or reference | Identifiers | Additional information |
|---|---|---|---|---|
| Genetic reagent (*M. musculus*) | B6.Cg-Tg(*Pdgfrb*-CreERT2)6,096Rha/J | Jackson Laboratory | #029684 RRID:IMSR_JAX:029684 | Pdgfrb-CreERT2 |
| Genetic reagent (*M. musculus*) | Ai96 | Jackson Laboratory | #028866 RRID:IMSR_JAX:024106 | GCaMP6s |
| Recombinant DNA reagent | AAV2-hSYN-hM3D (Gq)-mCherry | VVF, UZH Zuerich | ID: v101 | |
| Chemical compound, drug | Tamoxifen | Sigma-Aldrich | Cat. #: T5648 | 10 mg/ml |
| Chemical compound, drug | Isoflurane | Piramal Healthcare | Attane | |
| Chemical compound, drug | Texas Red Dextrane 70 kDa | Life Technologies | Cat. #: D-1864 | 2.5% |
| Chemical compound, drug | Nimodipine | Tocris | Cat. #: 0600 | |
| Chemical compound, drug | Endothelin-1 | Tocris | Cat. #: 1,160 | |
| Chemical compound, drug | TTX citrate | Tocris | Cat. #: 1,069 | |
| Chemical compound, drug | SKF96365 HCl | Tocris | Cat. #: 1,147 | |
| Chemical compound, drug | PNU-37883A HCl | Tocris | Cat. #: 2095 | |
| Chemical compound, drug | U46619 | Tocris | Cat. #: 1932 | |
| Chemical compound, drug | Clozapine | Tocris | Cat. #: 0444 | |
| Chemical compound, drug | UDP-Glucose 2Na+ | Sigma-Aldrich | Cat. #: 94,335 | |
| Chemical compound, drug | ATP | Sigma-Aldrich | Cat. #: A2383 | |
| Chemical compound, drug | DMSO | Sigma-Aldrich | Cat. #: 41,640 | |
| Software, algorithm | R studio | RStudio Team (2020) | v.1.0.136 | http://www.rstudio.com/ |
| Software, algorithm | GraphPad Prism | GraphPad Software La Jolla | V7.0 RRID:SCR_002798 | |
| Software, algorithm | Matlab | MathWorks | R2017b RRID:SCR_001622 | |

*Continued on next page*

*Continued*

| Reagent type (species) or resource | Designation | Source or reference | Identifiers | Additional information |
|---|---|---|---|---|
| Software, algorithm | Matlab code CHIPS | https://ein-lab.github.io/ *Barrett et al., 2018* | | |
| Software, algorithm | ImageJ | http://imageJ.nih.gov/ij | 1.53 c RRID:SCR_003070 | |
| Software, algorithm | ImageJ Vessel diameter plugin | PMID:20406650 | v.1.0 | |
| Software, algorithm | ScanImage | Janelia Research Campus | R3.8.1 RRID:SCR_014307 | |
| Other | Inline heated perfusion cube | ALA Scientific Instruments | HPC-G | |
| Other | Sapphire glass | UQG Optics | | Ø 3 × 3 mm |

## Animals

All animal experiments were approved by the local Cantonal Veterinary Office in Zürich (license ZH 169/17) and conformed to the guidelines of the Swiss Animal Protection Law, Swiss Veterinary Office, Canton of Zürich (Animal Welfare Act of 16 December 2005 and Animal Protection Ordinance of 23 April 2008). The following mice were interbred: B6.Cg-Tg(*Pdgfrb*-CreERT2)6,096Rha/J (*Pdgfrb*-CreERT2), The Jackson Laboratory (Stock #029684, *Gerl et al., 2015*) with Rosa26< LSL-GCaMP6s> (Ai96), The Jackson Laboratory (Stock #028866). Male and female offspring aged 2–9 months were used for experiments. The mice were given free access to water and food and were maintained under an inverted 12/12 hr light/dark cycle.

## Experimental timeline

To induce GCaMP6s expression, Tamoxifen (Sigma-Aldrich, cat. no. T5648), dissolved in corn oil (10 mg/ml), was injected intraperitoneally in mice at a dose of 100 mg/kg on four consecutive days, 3 weeks before ex vivo or in vivo imaging. After cranial window implantation, mice were allowed to recover for 3 weeks prior to in vivo imaging, to ensure that all surgery-related inflammation had resolved.

## Anesthesia

For surgery, animals were anesthetized with a mixture of fentanyl (0.05 mg/kg bodyweight; Sintenyl; Sintetica), midazolam (5 mg/kg bodyweight; Dormicum, Roche), and medetomidine (0.5 mg/kg bodyweight; Domitor; Orion Pharma), administered intraperitoneally. Anesthesia was maintained with midazolam (5 mg/kg bodyweight), injected subcutaneously 50 min after induction. To prevent hypoxemia, a face mask provided 300 ml/min of 100% oxygen. During two-photon imaging mice were anesthetized with 1.2% isoflurane (Attane; Piramal Healthcare, India) and supplied with 300 ml/min of 100% oxygen. Core temperature was kept constant at 37°C using a homeothermic heating blanket system (Harvard Apparatus) during all surgical and experimental procedures. The animal's head was fixed in a stereotaxic frame (Kopf Instruments) and the eyes were kept moist with ointment (vitamin A eye cream; Bausch & Lomb).

## Head-post implantation

A bonding agent (Gluma Comfort Bond; Heraeus Kulzer) was applied to the cleaned skull and polymerized with a handheld blue light source (600 mW/cm²; Demetron LC). A custom-made aluminum head-post was connected with dental cement (Synergy D6 Flow; Coltene/Whaledent AG) to the bonding agent on the skull for later reproducible animal fixation in the microscope setup. The skin lesion was treated with antibiotic ointment (Neomycin, Cicatrex; Janssen-Cilag AG) and closed with acrylic glue (Histoacryl; B. Braun). After surgery, the animals were kept warm and were given analgesics (Temgesic [buprenorphine] 0.1 mg/kg bodyweight; Sintetica).

## Virus injection and cranial window surgery

A 4 × 4 mm craniotomy was performed above the somatosensory cortex using a dental drill (Bien-Air Dental), and for experiments requiring chemogenetics, adeno-associated virus (AAV) vectors were

injected into the primary somatosensory cortex to achieve a localized chemogenetic receptor protein expression: 50 nl of AAV2-hSYN-hM3D (Gq)-mCherry (titer $1.02 \times 10^{11}$ VG/ml; Viral Vector Core Facility [VVF], University of Zürich) at a cortical depth of 300 µm. Large vessels were avoided to prevent bleeding. A square Sapphire glass coverslip (3 × 3 mm, UQG Optics) was placed on the exposed dura mater and fixed to the skull with dental cement, according to published protocols (*Holtmaat et al., 2009*).

## Behavior training for awake two-photon imaging

One week post-surgery, animals were handled multiple times a day for a week in order to get familiarized with the experimenter. Then, animals were adapted to the head fixation box by restraining them via the implanted head-post several times a day, with a gradual increase in restraint from seconds up to several minutes. After an extensive training period of 2–3 weeks, animals learned to sit still for the duration of a 45 min imaging session. Prior to an imaging session, 100 µl of a 2.5%, 70 kDa Texas Red Dextran (Life Technologies, cat. no. D-1864) was injected intravenously (iv.) into the tail vein to stain blood plasma. Mice were then let to recover from the iv. application and anesthesia for at least 1 hr before awake imaging was conducted.

## Two-photon imaging

Two-photon imaging was performed using a custom-built two-photon laser scanning microscope (2PLSM) (*Mayrhofer et al., 2015*) with a tunable pulsed laser (MaiTai HP system, Spectra-Physics and Chameleon Discovery TPC, Coherent Inc) and equipped with a 20× (W Plan-Apochromat 20 x/1.0 NA, Zeiss) or 25× (W Plan-Apochromat 25 x/1.05 NA, Olympus) water-immersion objective. During in vivo measurements, the animals were head-fixed and kept under anesthesia as described above. To visualize the vasculature Texas Red Dextran (2.5% w/v, 70,000 mw, 50 µl, Life Technologies, cat. no. D-1864) was injected intravenously. GCaMP6s and Texas Red were excited at 940 nm, and emission was detected with GaAsP photomultiplier modules (Hamamatsu Photonics) fitted with 535/50 nm and 607/70 nm band-pass filters and separated by a 560 nm dichroic mirror (BrightLine; Semrock). Control of microscope laser scanning was achieved with a customized version of ScanImage (r3.8.1; Janelia Research Campus; *Pologruto et al., 2003*).

At the beginning of each imaging session, z-stacks of the area of interest were recorded to identify the branch order of individual capillary segments. Once capillaries with a branch order of >4 were identified, a high resolution (512 × 512 pixels, 0.74 Hz) image was collected for reference and then baseline images (128 × 128 pixels; 11.84 Hz) were collected over a 90 s period with zoom factors ranging from 10 to 19. Multiple imaging sessions were conducted on different days for each animal. Calcium imaging of SMCs and EPs was limited to 25 min after isoflurane anesthesia induction.

## Acute brain slice preparation

Prior to slicing, 100 µl of a 2.5%, 70 kDa Texas Red Dextran (Life Technologies, cat. no. D-1864) was injected intravenously into the tail vein to stain blood plasma. Mice were euthanized by decapitation after deep anesthesia with isoflurane. The brain was extracted from the skull in ice-cold cutting solution consisting of 65 mM NaCl, 2.5 mM KCl, 0.5 mM $CaCl_2$, 7 mM $MgCl_2$, 25 mM glucose, 105 mM sucrose, 25 mM $NaHCO_3$, and 2.5 mM $Na_2HPO_4$. Slices of 300 µm thickness were cut using a Microm HM 650 V vibratome. Slices were immediately transferred into artificial cerebrospinal fluid (aCSF) consisting of 126 mM NaCl, 3 mM KCl, 2 mM $CaCl_2$, 2 mM $MgCl_2$, 25 mM glucose, 26 mM $NaHCO_3$, and 1.25 mM $NaH_2PO_4$, at 34°C. After a recovery phase of 1 hr, slices were used for imaging. All solutions were continuously gassed with 95% $O_2$, 5% $CO_2$ and were prepared on the day of experiment.

## Pharmacology in brain slices

Slices were imaged at 34°C in the same aCSF that was used for recovery after cutting. Solutions were infused by gravity into the slice chamber at a rate of 1.5 ml/min with an In-line Heated Perfusion Cube (HPC-G; ALA Scientific Instruments).

Nimodipine (Cat. No. 0600), U46619 (Cat. No. 1932), Endothelin-1 (Cat. No. 1160), TTX citrate (Cat. No. 1069), SKF96365 (Cat. No. 1147), and PNU-37883A HCl (Cat. No. 2095) were obtained from Tocris Bioscience (USA). UDP-Glucose (Cat. No. 94335), ATP (Cat. No. A2383), and all salts were obtained from Sigma-Aldrich.

Nimodipine, SKF96365, and PNU-37883A HCl were dissolved in DMSO, and Endothelin-1, UDP-Glucose, ATP and TTX citrate were dissolved in water as stock solutions at the highest possible molarity, stored at –20°C and were diluted to the desired concentration in aCSF prior to use.

For aCSF with increased potassium concentrations, the sodium-ion quantity was adapted to maintain the osmolarity of the solution constant.

## Acute slice electrophysiology

Coronal slices were transferred to a submerged recording chamber (RC-26, Warner Instruments, Hamden, CT) mounted on an upright Olympus microscope (BX61WI) equipped with differential interference contrast. aCSF was perfused and maintained at 34°C by an inline solution heater (TC-344C, Warner Instruments). Cells were visualized using a 40× water-immersion objective (LUMPlanFI/IR 40 x/0.80 W, Olympus). Somatic whole-cell recordings were performed from cortical layer 2/3 excitatory neurons using 3–4 MΩ glass pipettes filled with (in mM): 135 K-gluconate, 4 KCl, 2 NaCl, 10 HEPES, 4 EGTA, 4 Mg-ATP, 0.3 Na-GTP (pH 7.2, adjusted with KOH). Intrinsic properties of the neurons were measured in current clamp mode (in the absence of synaptic blockers) and depolarizing current injection steps of 500 ms duration were used to elicit action potentials. Tetrodotoxin TTX (1 µM) was added in the bathing solution to block neuronal spiking activity. Recordings were acquired with a Multiclamp 700B amplifier (Axon Instruments, Union City, CA), low-pass filtered at 2 kHz, digitized at 20 kHz (using Digidata 1,550B, Axon Instruments), and stored to disk using pClamp10 software (Molecular Devices, Sunnyvale, CA). Data analysis was performed offline using Clampfit 10.6 software (Molecular Device).

## Hypoxia pprotocol

To induce an acute hypoxic insult to the mice, the oxygen supply was transiently replaced with 2.0 l/min 100% nitrogen for 45 s.

## DREADD activation

To visualize and assess the area of hM3D(Gq)-mCherry transduced neurons, the laser wavelength was changed to 990 nm for overview imaging. $Ca^{2+}$ recordings were performed at a wavelength of 940 nm. hM3D(Gq) DREADD transduced neurons were activated with 30 µg/kg bodyweight clozapine (Tocris, Cat. No. 0444) (*Jendryka et al., 2019*) in saline (0.9% w/v) via a tail vein injection during image acquisition.

## Quantification and statistical analysis

Image analysis was performed using ImageJ and a custom-designed image processing toolbox, Cellular and Hemodynamic Image Processing Suite (CHIPS, *Barrett et al., 2018*), based on MATLAB (R2017b, MathWorks). For each field of view, all images were spectrally unmixed to reduce potential bleed-through between imaging channels and aligned using a 2D convolution engine to account for motion and x–y drift in time. Background noise was defined as the bottom first percentile pixel value in each frame and was subtracted from every pixel. Regions of interest (ROIs) were selected by combining two distinct methods in CHIPS: hand-selection of cell bodies as well as whole cell, and automated ROI detection with an activity-based algorithm (*Ellefsen et al., 2014*), both using anatomical images (128 × 128 pixel). A 2D spatial Gaussian filter ($\sigma_{xy}$ = 2 µm) and a temporal moving average filter (width = 1 s) were applied to all images to reduce noise. A moving threshold for each pixel was defined in the filtered stack as the mean intensity plus seven times the standard deviation of the same pixel during the preceding 2.46 s. Using this sliding box-car approach, active pixels were identified as those that exceeded the threshold. Active pixels were grouped in space (radius = 2 µm) and time (width = 1 s). Resulting ROIs with an area smaller than 4 µm² were considered to be noise and were excluded. We then combined the previously hand-selected ROIs into a single mask by subtracting the soma ROI from the whole cell territory ROIs, thereby leaving a mask of pericytes without soma. We then multiplied this 2D mask with each frame of the 3D mask obtained from the automated ROI detection in order to obtain a mask of ROIs within the cell territory and outside of the soma. We then extracted traces from two sets of ROIs for each image: the hand-selected soma ROIs and the adjusted 3D activity mask. The minimum distance from each activity ROI to the nearest soma was defined as the shortest distance between ROI edges. The signal vector (df/f) from each ROI was calculated using the mean intensity between 0.4 s and 4 s as baseline. Short, fast peaks were identified by applying a

digital band-pass filter with passband frequencies (f1 = 0.025 Hz and f2 = 0.2 Hz) before running the MATLAB findpeaks function. Noise peaks, due to motion or inflow of high-fluorescent particles, were manually removed.

Line scan acquisitions for vessel diameter were also analyzed using the custom-designed image processing toolbox for MATLAB, employing implemented methods described earlier (*Kim et al., 2012*; *Drew et al., 2010*; *Gao and Drew, 2014*). Vessel diameters for clozapine and hypoxia images were determined from time-averaged (20 s) stacks using the ImageJ vessel diameter plugin (*Fischer et al., 2010*).

Statistics for in vivo data was performed in RStudio (version 1.0.136) using the lme4 package for linear mixed-effects models (*Bates et al., 2015*). For fixed effects, we used the experimental condition (with/without drug) or cell type. For random effects, we had intercepts for individual animals and cells. Likelihood ratio tests comparing models with fixed effects against models without fixed effects were used to determine the model with the best fit while accounting for the different degrees of freedom. Visual inspection of residual plots did not reveal any obvious deviations from homoscedasticity or normality. All data were reported and plotted as uncorrected means. Frequencies are reported as signals/min ± SD, amplitudes as df/f ± SD and durations as s ± SD. p-values for different parameter comparisons were obtained using the multcomp package with Tukey's post hoc tests.

Graphics and statistical analyses of ex vivo experiments were performed using GraphPad Prism (version 7.00; GraphPad Software, La Jolla, CA). Datasets were tested with a Shapiro–Wilk normality test for Gaussian distribution. If one or both of the datasets passed the normality test, a two-tailed paired t-test was carried out; otherwise, a Wilcoxon matched-pairs signed rank test was chosen. For multiple comparisons, a one-way ANOVA was performed, using the Greenhouse–Geisser correction for sphericity and Tukey's post hoc tests for group comparisons. All statistical tests used to evaluate significance are indicated in the Figure legends along with the p-values. Values for area under the curve (auc) were calculated from 20 s windows of normalized traces for both baseline and drug responses considering the whole cell (soma and processes). For all in vivo and ex vivo experiments, N gives the number of animals and n is the number of cells. Statistical significances are highlighted based on: *p<0.05, **p<0.01, ***p<0.001.

## Data and Software Availability

The CHIPS toolbox for MATLAB is freely available on GitHub (https://ein-lab.github.io/; *Barrett et al., 2020*; *Barrett et al., 2018*). The source code for calcium signal detection with CHIPS is available as source code file.

## Acknowledgements

We thank the Viral Vector Facility of the University of Zürich for the supply of AAV vectors. We are grateful to Marc Zünd for assembly and maintenance of two-photon microscopes. Furthermore, we thank Zoe Looser for genotyping help. Karen Everett is thanked for critical proofreading of the manuscript. This project was supported by the Swiss National Science Foundation (Grant 310030_182703 and 31,003A_156965).

## Additional information

### Funding

| Funder | Grant reference number | Author |
|---|---|---|
| National Science Foundation | 310030_182703 | Chaim Glück<br>Annika Keller<br>Jillian L Stobart<br>Bruno Weber |
| National Science Foundation | 31003A_156965 | Chaim Glück<br>Annika Keller<br>Jillian L Stobart<br>Bruno Weber |

| Funder | Grant reference number | Author |
|--------|------------------------|--------|

The funder had no role in study design, data collection and interpretation, or the decision to submit the work for publication.

## Author contributions
Chaim Glück, Conceptualization, Data curation, Formal analysis, Investigation, Methodology, Project administration, Validation, Visualization, Writing – original draft, Writing – review and editing; Kim David Ferrari, Data curation, Formal analysis, Software, Validation, Writing – review and editing; Noemi Binini, Investigation, Validation, Writing – review and editing; Annika Keller, Conceptualization, Validation, Writing – review and editing; Aiman S Saab, Conceptualization, Formal analysis, Validation, Writing – review and editing, Supervision, Writing – original draft; Jillian L Stobart, Conceptualization, Formal analysis, Software, Validation, Writing – review and editing; Bruno Weber, Conceptualization, Funding acquisition, Resources, Supervision, Validation, Writing – review and editing

## Author ORCIDs
Chaim Glück ⓘ http://orcid.org/0000-0002-8754-9965
Kim David Ferrari ⓘ http://orcid.org/0000-0002-7565-1276
Annika Keller ⓘ http://orcid.org/0000-0003-1466-3633
Aiman S Saab ⓘ http://orcid.org/0000-0003-3886-8369
Bruno Weber ⓘ http://orcid.org/0000-0002-9089-0689

## Ethics
All animal experiments were approved by the local Cantonal Veterinary Office in Zürich (license ZH 169/17) and conformed to the guidelines of the Swiss Animal Protection Law, Swiss Veterinary Office, Canton of Zürich (Animal Welfare Act of 16 December 2005 and Animal Protection Ordinance of 23 April 2008). Every effort was made to minimize suffering and conform to the 3Rs principles.

## Decision letter and Author response
Decision letter https://doi.org/10.7554/eLife.70591.sa1
Author response https://doi.org/10.7554/eLife.70591.sa2

## Additional files

### Supplementary files
- Transparent reporting form
- Source code 1. Source Code file for CHIPS.

### Data availability
All data generated or analysed during this study are included in the manuscript and supporting files. Source data files have been provided for Figures 2, 3, 4, 5 and 6.

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
