## [Decision Letter]

**Acceptance summary:**

This is a rigorous and comprehensive study of brain vascular mural cell physiology using state of the art imaging , electrophysiology and pharmacology. This study will be of interest to neuroscientists and vascular biologists.

**Decision letter after peer review:**

[Editors’ note: the authors submitted for reconsideration following the decision after peer review. What follows is the decision letter after the first round of review.]

Thank you for submitting your work entitled "Distinct signatures of calcium activity in brain pericytes" for consideration by *eLife*. Your article has been reviewed by 3 peer reviewers, and the evaluation has been overseen by a Reviewing Editor and a Senior Editor. The following individuals involved in review of your submission have agreed to reveal their identity: Andy Shih (Reviewer #2); Jaime Grutzendler (Reviewer #3).

Our decision has been reached after excessive consultation between the reviewers. Based on these discussions and the individual reviews below, we regret to inform you that your work will not be considered further for publication in *eLife*.

The reviewers felt that the calcium signaling measurements in vivo were of significant interest. However, there were many technical concerns including the viability of the brain slice preparation and interpretation of potassium channel drug effects on multiple cell types. We felt that these issues would take considerable time, and would likely change the conclusions of the paper. *eLife* policy indicates that revisions should be limited to those that can be reasonably accomplished in two months. We would be willing to consider a new manuscript that addressed the concerns of the reviewers.

*Reviewer #1:*

Summary:

In their manuscript, Glück et al., seek to define Ca^2+^ signaling characteristics in brain pericytes, focusing on differences in Ca^2+^ events between arteriole-proximate ensheathing pericytes (EPs) and more distal pericytes, which they refer to as capillary pericytes (CPs). Using 2-photon imaging in an anesthetized mouse cranial window model, they report distinct differences in Ca^2+^ event frequency, amplitude and duration between EPs and CPs. They further describe a clear difference in the Ca^2+^-contraction relationship between these two different pericyte populations, reporting that changes in Ca^2+^ in EPs are rapidly followed by changes in vessel diameter, a response pattern that mirrors that in arteriolar smooth muscle cells (SMCs); in contrast, CPs showed no change in diameter in response to Ca^2+^ events. Subsequent experiments in acute brain slices suggested differences in Ca^2+^ signal persistence ex vivo versus in vivo and provided evidence that vasomodulating Gq-protein-coupled receptor (GqPCR) agonists are capable of driving Ca^2+^ events in CPs. Pharmacological interventions suggested that these events are dependent on Ca^2+^ influx via TRPC (transient receptor potential canonical) channels, but independent of ongoing neuronal activity. Their data further suggest that the L-type voltage-gated Ca^2+^ channel may also mediate Ca^2+^ influx, although this is not likely a major influx pathway as the selective blocker nimodipine had no effect on Ca^2+^ events in processes and only a very modest effect on such events in somata. Finally, on the basis of pharmacological approaches in brain slices, including DREADD-based experiments, direct application of K^+^ and ischemia/hypoxia-mimetic conditions (sodium azide/100% N2 exposure), they suggest that K^+^ released in response to neuronal stimulation causes a decrease in Ca^2+^ events in CPs, and that Kir and KATP channels mediate this effect.

General comments/questions

Data are dumped wholesale in parenthetic expressions together with statistical information and figure call-outs, making it difficult to assess relationships. Such data-rich sentences should be re-written to more closely juxtapose data and associated conditions for more facile comparison. For example (lines 201-203):

Before: "Comparing the signal frequency between in vivo and ex vivo, there was a more than twofold reduction in calcium activity in both somata and processes (in vivo, S: 7.8 (2.5) and P: 18.9 (9.4); ex vivo, S: 3.1 (1.6) and P: 7.3 (5.6). Values are mean (SD) signals/min. Figure 3A)."

After: "A comparison of calcium activity showed that calcium signal frequency, expressed as mean ± standard deviation (SD), was reduced more than 2-fold ex vivo compared with that in vivo in both processes (7.3 ± 5.6 vs. 18.9 ± 9.4) and somata (3.1 ± 1.6 vs. 7.8 ± 2.5) (Figure 3A)."

Distinct basal calcium transients of mural cells in vivo

Lines 126-7: "Basal calcium signals in SMCs and EPs 126 were usually only visible during the first 30 minutes of anesthesia." When exactly during this 30-min period were Ca^2+^ events in EPs recorded? Immediately after delivering anesthesia (i.e., before anesthesia effects on signaling manifested)? Some time during the 30-minute period? At the end of the 30-minute period? Measurements made immediately after delivering anesthesia, though more difficult to obtain, might provide physiologically meaningful information, but the only thing learned from responses measured during or at the end of the 30-minute period is how anesthesia affects Ca^2+^ signaling in EPs.

Persisting calcium signals in CPs ex vivo

Line 204-5: In many cases (Figure 3B), arteries and arterioles were either collapsed due to loss of tone or intraluminal pressure in the slice preparation or were constricted by possibly dead SMCs or EPs (no detectable calcium signal). These observations are disturbing signs of possible tissue injury and raise questions about the validity of data obtained using acute brain slice preparations. Faced with these observations, the authors should confirm cell viability in brain slice preparations after all pharmacological interventions that reduce Ca^2+^ signaling (e.g., hM3D(Gq)-DREADD/clozapine, SKF) to rule out the possibility that observed decreases in Ca^2+^ events reflect the trivial case of signal rundown in dying cells.

Line 226-7: "…massive calcium response in CPs when U46619 was included in the superfusate (Figure 3D, 226 Video 5). This overt calcium response in CPs was accompanied by cytoplasmic extrusions, suggesting that U46619 might be toxic for pericytes." Although U46610 can have harsh effects, including pericyte blebbing, the severity of U46610 effects described here suggest that procedures used to isolate, prepare and/or maintain brain slices increase the vulnerability of the preparation to potentially toxic stimuli.

Line 233: Header for Figure 3 legend (Calcium dynamics in the absence of blood flow ex vivo is more affected in EPs than CPs) implies that differences in EP Ca^2+^ dynamics between the in vivo and ex vivo setting is caused by the absence of blood flow in the latter preparation. This is an overstatement; no data are provided to support a direct relationship between blood flow and EP Ca^2+^ dynamics.

Figure 3 (all): Are these all z-stacks/z-projections of vessels? If so, this should be indicated in the figure and/or figure legend. The appearance of pinching or narrowing could be due to vessel orientation relative to the imaging plane.

CP calcium events are evoked by vasomodulators

Line 279: "Application of Nimodipine (100 μM)…" What is the rationale for using 100 μM nimodipine? This is an astonishingly high concentration for a dihydropyridine-even in a brain slice. A rationale is similarly required for SKF, which is also normally used at considerably lower concentrations than 100 μM.

Lines 288-9: "We applied tetrodotoxin (TTX), a voltage-gated sodium channel blocker, to dampen neuronal activity (Zonta et al., 2003)….CP calcium signals were not affected" The absence of an effect of TTX could simply indicate that cells in the brain slice preparation are unresponsive because they are dead or dying. The viability of cells in the prep needs to be verified.

Potassium released by neuronal stimulation leads to a calcium signal drop in CPs

Lines 322-4: "We used chemogenetics (Roth, 2016), in which neurons – in this case transduced to express hM3D(Gq)-DREADD – were activated with 30 μg/kg iv. clozapine." Transduction procedure needs to be described in Methods, and possible toxicity associated with it needs to be tested by assessing cell viability before and after transduction. In the absence of these latter control experiments, the possibility that the loss of Ca^2+^ signaling results from toxic effects of clozapine or toxicity of the transduction procedure itself cannot be ruled out.

Lines 332-338: Based on the set up for this section ("potassium is sensed by SMCs, causing the suppression of calcium oscillations"), the authors seem to be operating under the assumption that 10 mM K^+^ and BaCl2 are acting directly on mural cells-CPs in the current context. This is an incredibly bold assumption given that effects of 10 mM K^+^ and BaCl2 could be indirect through hyperpolarization of endothelial cells, which express Kir2.1 channels that are robustly activated by extracellular K^+^ and are blocked by 100 μM BaCl2.

Effects of KATP channel blockers could similarly be indirect through actions on endothelial cells. Although RNA seq data would suggest a large role for these channels in pericytes, further confirmation of their function should be provided, such as activation with pinacidil and block of the KATP channel during metabolic changes.

Overall

Using predominantly an ex vivo acute brain slice preparation, the authors report differences in Ca^2+^ signaling signatures between EPs and CPs, lending additional experimental weight to the growing body of evidence that arteriole-proximate and more distal pericytes are functionally distinct. They describe differential effects of anesthesia on basal Ca^2+^ signaling in EPs (gradually eliminated) and CPs (largely unaffected) and show that Ca^2+^ signaling in EPs is reduced in brain slices. They further intimate (e.g., see Abstract, Figure 3 legend) that Ca^2+^ signaling in EPs requires blood flow; however, their data do not support this conclusion since flow effects on Ca^2+^ events were not tested directly. Importantly, the viability of cells in brain slices was not assessed following pharmacological manipulations that decreased Ca^2+^ signaling. In the absence of such control experiments, it is not possible to rule out tissue injury or improper slice maintenance as the cause of such decreases, especially given evidence for dead SMCs and EPs and unusually severe effects of U46619. Accordingly, some of the more compelling data from ex vivo pharmacological manipulations are suspect. The transient nature of EP Ca^2+^ responses recorded by 2-photon microscopy in cranial window model mice, reflecting signal dampening associated with the onset of anesthesia, similarly cast doubt on the significance of these in vivo findings. None of their data, including chemogenetic activation of hM3D(Gq)-DREADD-expressing neurons and pharmacological manipulations involving K^+^ and BaCl2, support the specific conclusion that neuronal K^+^ release causes the decrease/loss of pericyte calcium signaling. Importantly, pharmacological manipulations of Kir2 and KATP channels do not exclude endothelial cell contributions-especially given the importance of endothelial cell Kir2 in neurovascular coupling. More generally, the interconnected electrical properties of the microvasculature are not discussed or accounted for within the overall experimental design.

*Reviewer #2:*

This is an exciting and timely study to uncover the mechanisms and physiological relevance of pericyte calcium signaling. By using pericyte specific calcium sensing-reporter mice (Pdgfrbeta-CreERT2; GCaMP6s (Ai96-flox)), Gluck et al., identify distinct Ca signatures within the heterogeneous populations of pericytes along the brain vasculature including smooth muscle cells (SMCs), ensheathing pericytes (EPs; found along the branch orders 1-4), and capillary pericytes (CPs; found along > 4th order). While SMCs and EPs have synchronous calcium oscillations that are expectedly related to vessel diameter changes, CPs have asynchronous calcium signaling in somata and processes and there is no correlation with vessel diameter changes in the healthy anesthetized animal. This has been shown in some prior studies, but the current manuscript delves far deeper into the basis of pericyte calcium signals.

To pharmacologically test how calcium signals are regulated in pericytes, they moved to ex vivo, brain slice experiments. They show that blocking VGCC and TRPC reduces CP calcium signals. Activating neurons using chemogenetics reduced CP calcium signals in vivo. KCl, to mimic neuronal activity, also reduced CP calcium and this effect is attenuated by blocking Kir2 and Kir6.1 ex vivo. Creating hypoxia-like conditions (in vivo: N2 or ex vivo: sodium azide) reduced calcium signals in CPs. Conversely, a number of vasoactive substances known to constrict arterioles lead to large calcium influxes in CPs, including ET-1, ATP and UDP-glucose.

These studies pave new roads to understanding how pericytes are involved in NV coupling, and blood flow control. The approach is highly innovative and the in vivo-ex vivo approach is a major strength. However, there is a missing link that makes it difficult to understand the physiological relevance of the CP calcium signals as currently presented (comments 1 and 2). If addressed, this would substantially increase the impact of the work.

1) It is not clear how calcium signaling in CPs is related to electrical conductance along the capillaries leading to arteriole diameter change. For example, how do CP calcium changes relate to calcium in SMCs and EPs (and/or arteriolar diameter) on the same vascular tree during NV coupling in vivo? How does CP calcium levels relate to endothelial activity? Is it possible to block CP K^+^ signaling in vivo to test the hypothesis that upstream arteriolar dilation will be decreased? Perhaps the relatively selective expression of KATP in CPs can be used to leverage an in vivo pharmacological experiment. It is not necessary to address all these questions, but some additional data to link CP calcium to arteriolar dilation would strengthen the paper and substantiate the compelling model in Figure 6.

2) Local regulation of capillary flow: The presumption is that the sustained calcium changes evoked by vasomediators (ET-1, ATP) and neural activity are affecting local capillary diameter. This would be helpful to know because past studies have not been as rigorous in reporting vascular branch orders, leaving some ambiguity as the authors note. The data would help us understand whether CPs are autonomously able to regulate capillary diameter. This is important even if the stimulation becomes supra-physiological, as it clarifies whether CPs are a logical target for constriction in pathologies such as stroke, AD. Curiously, some of the data also show a slower component of calcium modulation can occur on the scale of tens of second to minutes (traces of Figure 5). Are these slow modulations related to capillary diameter? Similarly, how does N2-induced hypoxia result in vascular diameter changes along the various pericyte territories in relation to Ca signal changes?

3) Neural activity: DREADDs are gated by clozapine-N-oxide, rather than clozapine. It is correct that clozapine was used in this study, instead of its less bioactive counterpart? If so, a control non-DREADD expressing group might be needed to verify the intended goal of stimulating local neurons in vivo, as opposed to broad actions of clozapine. Further, it is suggested that basal neuronal activity in slice does not regulate CP calcium fluctuations because TTX has no effect. However, neuronal activity was not actually measured.

4) In addition to a Non-DREADD expressing control, some additional controls would increase rigor. To ensure that the fluctuation in calcium dynamics in pericytes is not influenced by animal motion or passing RBCs in vivo, comparison to a general fluorescent reporter like GFP would be helpful. Vehicle control experiments are not described for the ex vivo pharmacological experiments. If these were performed, it would be good to show.

*Reviewer #3:*

The manuscript by Glück et al., examined the patterns of calcium transients in vascular mural cells in the live mouse brain and brain slices using two photon imaging of calcium sensor mice, in combination with pharmacological and chemogenetic manipulations. They found that pericytes exhibit distinct calcium dynamics compared to vascular smooth muscle cells (vSMCs). Pericytes in vivo have compartmentalized, irregular, higher-frequency calcium transients, which are not correlated with vessel diameter changes, whereas calcium fluctuation of vSMCs are highly correlated with changes in vessel diameter. In ex vivo preparations, pericytes retained their spontaneous calcium properties in contrast to its loss in SMCs. Combined with chemogenetics in vivo and pharmacological manipulation in slices, the authors demonstrated neuronal activity can decrease calcium response in pericytes by elevating extracellular potassium, which is mediated by Kir2.2, Kir6.1. Energetics state, and hypoxia challenge can also affect pericyte Ca^2+^ activation through KATP channel.

Overall, this paper is technically rigorous and reproduces the majority of in vivo findings by recent publications (PMIDs: 26119027, 29937277). It also provides a more granular investigation of Ca^2+^ transients in pericytes (comparing soma and processes) and the pharmacological investigation of potassium channels and ATP in slices is also novel.

1) The main criticism we have about this paper is the nomenclature used to define the different mural cells which continues to confuse the field. Why use the title "signatures of calcium in brain pericytes" instead of simply saying in mural cells? This is surprising given that their data completely parallels data showing that SMCs and pericytes are distinct cells types. They introduce again the confusing term "ensheathing pericytes" despite their finding that EP essentially have identically characteristics to vSMCs (near identical calcium properties, contractility, expression of aSMA, circumferential anatomy, lack of compartmentalized Ca^2+^ signals and others) as well as findings from other groups including transcriptome data showing no such thing as subpopulations of pericytes (PMIDs: 29443965, 30129931) and data from uptake of small fluorescent molecules specifically by CP but not other mural cells (PMID: 28504673). We think it would be critical for this paper to clarify the field rather than continue to confuse the nomenclature and cite historical 100 old anatomical studies to justify it. Otherwise it is essential that they produce functional, structural and genetic evidence that there is such thing as a distinct ensheathing cell that is closer to pericytes than to vSMCs to justify the name. The logical terminology would be to call them "terminal vSMCs" or something similar to account for the minor morphological difference these cells have compared to the slightly more proximal vSMCs.

2) Although not essential for this project, it would have been nice to obtain concurrent endothelial and pericyte Ca recordings (using RCaMP and GCaMP sensors) to really understand the role of CP Ca^2+^. Is it similar to the described hyperpolarization propagation in endothelium? (PMID: 28319610), are there any interactions between CP and endothelium?

3) They should also comment on the fact that it is very difficult to target pericytes without having impact on other cell types by pharmacological manipulation. Only Kir6.6 (Kcnj8) blocking provides pericyte specificity, since BaCl2 can also affect endothelial Kir channels (PMID: 26840527), which may indirectly change pericytes calcium.

4) Is there any directionality in the propagation of the Ca transients in pericytes in vivo?

5) Do the microdomains of Ca in vitro become more synchronized between processes and Soma. If so, could this mean that in the in vitro prep the microenvironment around pericytes is more homogeneous than in vivo, thereby causing more homogeneity of Ca?

6) In their figure 6 model there is no mention of possible pericyte to pericyte or pericyte to SMC communication. This would be worth considering as these cells are likely to be gap junction coupled.

---

## [Author Response]

[Editors’ note: the authors resubmitted a revised version of the paper for consideration. What follows is the authors’ response to the first round of review.]

Reviewer #1:Summary:In their manuscript, Glück et al., seek to define Ca^2+^ signaling characteristics in brain pericytes, focusing on differences in Ca^2+^ events between arteriole-proximate ensheathing pericytes (EPs) and more distal pericytes, which they refer to as capillary pericytes (CPs). Using 2-photon imaging in an anesthetized mouse cranial window model, they report distinct differences in Ca^2+^ event frequency, amplitude and duration between EPs and CPs. They further describe a clear difference in the Ca^2+^-contraction relationship between these two different pericyte populations, reporting that changes in Ca^2+^ in EPs are rapidly followed by changes in vessel diameter, a response pattern that mirrors that in arteriolar smooth muscle cells (SMCs); in contrast, CPs showed no change in diameter in response to Ca^2+^ events. Subsequent experiments in acute brain slices suggested differences in Ca^2+^ signal persistence ex vivo versus in vivo and provided evidence that vasomodulating Gq-protein-coupled receptor (GqPCR) agonists are capable of driving Ca^2+^ events in CPs. Pharmacological interventions suggested that these events are dependent on Ca^2+^ influx via TRPC (transient receptor potential canonical) channels, but independent of ongoing neuronal activity. Their data further suggest that the L-type voltage-gated Ca^2+^ channel may also mediate Ca^2+^ influx, although this is not likely a major influx pathway as the selective blocker nimodipine had no effect on Ca^2+^ events in processes and only a very modest effect on such events in somata. Finally, on the basis of pharmacological approaches in brain slices, including DREADD-based experiments, direct application of K^+^ and ischemia/hypoxia-mimetic conditions (sodium azide/100% N2 exposure), they suggest that K^+^ released in response to neuronal stimulation causes a decrease in Ca^2+^ events in CPs, and that Kir and KATP channels mediate this effect.

We thank the reviewer for the constructive feedback. We have revised our manuscript with respect to the referees’ comments. We have performed several additional experiments and analyses to address the concerns raised which helped to strengthen the conclusions of the study.

General comments/questionsData are dumped wholesale in parenthetic expressions together with statistical information and figure call-outs, making it difficult to assess relationships. Such data-rich sentences should be re-written to more closely juxtapose data and associated conditions for more facile comparison. For example (lines 201-203):Before: "Comparing the signal frequency between in vivo and ex vivo, there was a more than twofold reduction in calcium activity in both somata and processes (in vivo, S: 7.8 (2.5) and P: 18.9 (9.4); ex vivo, S: 3.1 (1.6) and P: 7.3 (5.6). Values are mean (SD) signals/min. Figure 3A)."After: "A comparison of calcium activity showed that calcium signal frequency, expressed as mean ± standard deviation (SD), was reduced more than 2-fold ex vivo compared with that in vivo in both processes (7.3 ± 5.6 vs. 18.9 ± 9.4) and somata (3.1 ± 1.6 vs. 7.8 ± 2.5) (Figure 3A)."

Thank you for pointing this out. We have re-written data-rich sentences to facilitate the reading and comparisons of statistical information.

*Distinct basal calcium transients of mural cells* in vivo

Lines 126-7: "Basal calcium signals in SMCs and EPs 126 were usually only visible during the first 30 minutes of anesthesia." When exactly during this 30-min period were Ca^2+^ events in EPs recorded? Immediately after delivering anesthesia (i.e., before anesthesia effects on signaling manifested)? Some time during the 30-minute period? At the end of the 30-minute period? Measurements made immediately after delivering anesthesia, though more difficult to obtain, might provide physiologically meaningful information, but the only thing learned from responses measured during or at the end of the 30-minute period is how anesthesia affects Ca^2+^ signaling in EPs.

Ca^2+^ imaging data from SMCs and EPs were acquired within 25 min after the onset of isoflurane anesthesia. This is now clarified in the manuscript and in the method section.

Moreover, to rule out any possible anesthesia effects on the observed differences in Ca^2+^ dynamics between mural cells in vivo, we have now performed additional experiments in awake animals. The new data of awake imaging is now included in new Figure 2L-N. Importantly, although there were some differences in mural Ca^2+^ dynamics between anesthesia and awake imaging (see new Figure 2 —figure supplement 4), the differences between EPs and CPs regarding calcium signal frequency and duration were very consistent also in awake animals.

*Persisting calcium signals in CPs* ex vivo

Line 204-5: In many cases (Figure 3B), arteries and arterioles were either collapsed due to loss of tone or intraluminal pressure in the slice preparation or were constricted by possibly dead SMCs or EPs (no detectable calcium signal). These observations are disturbing signs of possible tissue injury and raise questions about the validity of data obtained using acute brain slice preparations. Faced with these observations, the authors should confirm cell viability in brain slice preparations after all pharmacological interventions that reduce Ca^2+^ signaling (e.g., hM3D(Gq)-DREADD/clozapine, SKF) to rule out the possibility that observed decreases in Ca^2+^ events reflect the trivial case of signal rundown in dying cells.

We agree with the reviewer, that assessment of cell viability in acute brain slices is crucial for proper data interpretation.

A loss of vascular tone by lack of blood pressure in the acute slice preparation is inevitable. However, in our study we aimed to complement our in vivo observations with acute slice experiments, which allows better access for pharmacological interventions e.g. to gain some mechanistic insights underlying the Ca^2+^ signals in pericytes.

We regret that our initial phrasing (“were constricted by possibly dead SMCs or EPs (no detectable calcium signal)”) was misleading. To better understand why SMCs and EPs showed strongly reduced Ca^2+^ signals in acute slice preparations, we have now performed additional experiments. Previous studies on SMCs and EPs in acute slice preparations were done in the presence of the widely used constricting agent U46619 (Mishra et al., 2014, Filosa et al., 2004). Strikingly, we revealed that initially “silent” SMCs and EPs regain Ca^2+^ activity upon bath-application of 100 nM U46619 (See new Figure 3E, F), thus showing that these cells are responsive and viable in our slice preparations. This also clarifies our previous observations of low to no Ca^2+^ signals in SMCs and EPs in the absence of vascular preconstruction (new Figure 3C, D). Incidentally, a recent study by the Nelson lab (Gonzales et al., 2020) confirms this finding in a retinal preparation.

Furthermore, to rule out that the reduction of Ca^2+^ signals observed in CPs by pharmacologic intervention with SKF (previous Figure 2E) is not simply a calcium run-down effect, we performed additional experiments with SKF and included an end-stimulus (addition of 100 nM U46619), to elicit a Ca^2+^ response (as shown in Figure 3D). We confirmed that SKF decreases Ca^2+^ activity in CPs, however, more importantly, this was not due to a run-down effect since CPs were viable and responsive, as shown by the large Ca^2+^ increase upon U46619 stimulation following SKF incubation (see new Figure 4E).

We would like to point out, that dying cells usually exhibit a sustained elevation of Ca^2+^ due to inactive ion-pumps and loss of membrane integrity (Carafoli and Krebs, 2016) and thus appear as very bright (Hill et al., 2017) and rounded cells, which we never observed during our experiments with pharmacological interventions. In the other cases where we observed a transient drop in Ca^2+^ activity, such as with sodium azide, hypoxia or potassium, Ca^2+^ signaling activity was always restored after washout, as shown in the traces (Figures 6A/D/E).

Furthermore, we want to clarify that all clozapine experiments were performed in vivo and not in acute slices. Nonetheless, we have now included new experiments to rule out possible clozapine induced side effects (see also further below):

1)We included Ca^2+^ recordings of CPs outside the virally transfected area of hM3D(Gq)-DREADD expressing neurons and these cells did not show any changes in Ca^2+^ signaling frequency upon clozapine treatment (see new Figure 5 —figure supplement 1).

2) Mural cells that showed a drop in Ca^2+^ signaling frequency in response to neuronal (DREADD) activation were monitored a day later and the same cells exhibited normal Ca^2+^ activity (new Figure 5 —figure supplement 2), clearly ruling out clozapine-induced cell death for the neuronal stimulation-evoked decrease in mural cell Ca^2+^ activity.

Line 226-7: "…massive calcium response in CPs when U46619 was included in the superfusate (Figure 3D, 226 Video 5). This overt calcium response in CPs was accompanied by cytoplasmic extrusions, suggesting that U46619 might be toxic for pericytes." Although U46610 can have harsh effects, including pericyte blebbing, the severity of U46610 effects described here suggest that procedures used to isolate, prepare and/or maintain brain slices increase the vulnerability of the preparation to potentially toxic stimuli.

We understand the reviewer’s concerns, however, we do not share the opinion that the U46619 effects observed were due to bad slice preparation or increased vulnerability of slices in our hands (see also previous comments regarding cell viability and patch clamp recordings of neurons further below).

We would like to point out that CPs were only stimulated by a short pulse of U46619. This led to a strong Ca^2+^ increase accompanied by changes in cellular morphology. However, since the cytosolic Ca^2+^ levels of the stimulated cells returned back to baseline after washout of U46619 (Figure 3D), we conclude that the cellular Ca^2+^ extrusion mechanisms were still intact. Further, the morphological changes of CPs were reversed after a prolonged washout of U46619(> 30 min) which we now added to (new Figure 3G, H).

Moreover, the morphological changes in U46619 stimulated CPs could hint to a cellular mechanism involving cytoskeletal rearrangements, as has been observed in an earlier study (Fernandez-Klett et al., 2010). Additionally, membranous blebs have been associated to cytoskeletal rearrangements (Robertson et al., 2021, Kondrychyn et al., 2020) resulting in changes in mechanical properties of endothelial cells.

Since U46619 has been shown to result in CP constriction (Fernandez-Klett et al., 2010), our observations on morphological changes of CPs suggest similar cytoskeletal rearrangements involved in movement/constriction. Moreover, also others recently reported the occurrence of membranous blebs along vasoconstriction in response to optogenetic stimulation of CPs (Hartmann et al., 2021).

We have discussed this possibility in the revised manuscript and added new figures to better showcase these transient morphological changes in CPs (see new Figure 3G, H and Figure 3 —figure supplement 2).

Line 233: Header for Figure 3 legend (Calcium dynamics in the absence of blood flow ex vivo is more affected in EPs than CPs) implies that differences in EP Ca^2+^ dynamics between the in vivo and ex vivo setting is caused by the absence of blood flow in the latter preparation. This is an overstatement; no data are provided to support a direct relationship between blood flow and EP Ca^2+^ dynamics.

We agree with the reviewer that this was an overstatement. We changed the header of Figure 3 as follows: “Mural cell calcium dynamics in acute cortical brain slices”.

Figure 3 (all): Are these all z-stacks/z-projections of vessels? If so, this should be indicated in the figure and/or figure legend. The appearance of pinching or narrowing could be due to vessel orientation relative to the imaging plane.

We thank the reviewer for pointing this out. Some images in Figure 3 were updated (also with the new data about SMC and EP Ca^2+^ activity). Images in Figure 3B and 3C are z projections and this has been now clarified in the figure legend.

CP calcium events are evoked by vasomodulatorsLine 279: "Application of Nimodipine (100 μM)…" What is the rationale for using 100 μM nimodipine? This is an astonishingly high concentration for a dihydropyridine-even in a brain slice. A rationale is similarly required for SKF, which is also normally used at considerably lower concentrations than 100 μM.

The higher concentrations for these drugs were chosen to reach sufficient inhibition. Performing a dose-dependent inhibition analysis on CP Ca^2+^ dynamics with nimodipine and SKF would be out of the scope for a revision.

Lines 288-9: "We applied tetrodotoxin (TTX), a voltage-gated sodium channel blocker, to dampen neuronal activity (Zonta et al., 2003)….CP calcium signals were not affected" The absence of an effect of TTX could simply indicate that cells in the brain slice preparation are unresponsive because they are dead or dying. The viability of cells in the prep needs to be verified.

Ca^2+^ signals in CPs were persisting and were not affected by addition of TTX, therefore we conclude that the observed cells were viable. Of note, dead cells would not show spontaneous activity (Wu et al., 2019).

However, to showcase that also in our slice preparations cortical neurons are viable and fire action potentials, we now added new data (new Figure 6 —figure supplement 1). As expected, action potential firing in L2/3 neurons was completely abolished by TTX. Moreover, intrinsic membrane properties of these neurons were comparable to previous reports (Oswald and Reyes, 2008, Brown et al., 2019, Avermann et al., 2012).

Potassium released by neuronal stimulation leads to a calcium signal drop in CPsLines 322-4: "We used chemogenetics (Roth, 2016), in which neurons – in this case transduced to express hM3D(Gq)-DREADD – were activated with 30 μg/kg iv. clozapine." Transduction procedure needs to be described in Methods, and possible toxicity associated with it needs to be tested by assessing cell viability before and after transduction. In the absence of these latter control experiments, the possibility that the loss of Ca^2+^ signaling results from toxic effects of clozapine or toxicity of the transduction procedure itself cannot be ruled out.

We understand the concerns of the reviewer, that AAV transduction could be toxic. Our lab has longstanding experience in using AAVs for in vivo expression of genetically encoded sensors without inducing neuronal cell death, reactive astrogliosis and microglia activation (Zuend et al., 2020, Stobart et al., 2018, Machler et al., 2016). If transduction would be toxic, there should be no Ca^2+^ signals detectable in mural cells, which was not the case.

We described the transduction procedure in the Materials and methods under section “Virus injection and cranial window implantation:

(Lines 704-708: “A 4 x 4 mm craniotomy was performed above the somatosensory cortex using a dental drill (Bien-Air Dental), and for experiments requiring chemogenetics, adeno-associated virus (AAV) vectors were injected into the primary somatosensory cortex to achieve a localized chemogenetic receptor protein expression: 50 nl of AAV2-hSYN-hM3D (Gq)-mCherry (titer 1.02 x 1011 VG/ml; viral vector core facility (VVF), University of Zürich) at a cortical depth of 300 µm”).

As mentioned in an earlier comment (see above, point 3), we now included new experiments to rule out possible clozapine-induced side effects or toxicity.

1) We included Ca^2+^ recordings of CPs outside the virally transduced area of hM3D(Gq)-DREADD expressing neurons and these cells did not show changes in Ca^2+^ signaling frequency upon clozapine treatment (see new Figure 5 —figure supplement 1).

2) Mural cells that showed a drop in Ca^2+^ signaling frequency in response to neuronal (DREADD) activation were monitored a day later and the same cells exhibited normal Ca^2+^ activity (new Figure 5 – figure supplement 2) ruling out clozapine-induced cell death for the neuronal stimulation evoked decrease in mural cell activity.

Lines 332-338: Based on the set up for this section ("potassium is sensed by SMCs, causing the suppression of calcium oscillations"), the authors seem to be operating under the assumption that 10 mM K^+^ and BaCl2 are acting directly on mural cells-CPs in the current context. This is an incredibly bold assumption given that effects of 10 mM K^+^ and BaCl2 could be indirect through hyperpolarization of endothelial cells, which express Kir2.1 channels that are robustly activated by extracellular K^+^ and are blocked by 100 μM BaCl2.Effects of KATP channel blockers could similarly be indirect through actions on endothelial cells. Although RNA seq data would suggest a large role for these channels in pericytes, further confirmation of their function should be provided, such as activation with pinacidil and block of the KATP channel during metabolic changes.

We agree with the reviewer that pharmacological experiments in acute brain slices are not celltype specific. Given the tight association of CPs with capillary endothelial cells there is a fundamental methodological problem to disentangle effects on either cell type. Indeed, both cell-types express the same channels that may influence cellular function in similar ways (e.g. see also a recent preprint of the Nelson lab reporting in retinal preparations functional expression of KATP channels in capillary endothelial cells and CPs (Sancho et al., 2021). Future studies employing more refined methods, such as cell-type specific knock-out animals or cell-type specific drug targeting could shed more light onto the intercellular relationship of CPs and capillary endothelial cells. However, at the moment there is no direct evidence of how changes in cortical endothelial cells impact CP behavior and vice versa.

Nonetheless, we are fully aware that we cannot base the effects of potassium, barium or KATP blockers solely on CPs, and we never intended to phrase it that way. We are aware of the effect of potassium and barium on endothelial cells and have acknowledged these studies several times in the manuscript. Whether the observed potassium induced Ca^2+^ signaling drop in CPs is a direct effect or via indirect actions involving capillary endothelial cells, cannot be properly disentangled at the moment.

To clarify these points, we have now changed the abstract, Results section and discussion and also included the possible cellular interaction between endothelial cells and CPs in our New summary Figure 7.

New discussion: “Potassium is released in high amounts during the repolarization phase after action potential firing (Paulson and Newman, 1987) and can act as a potent vasomodulator (McCarron and Halpern, 1990). Potassium sensing by SMCs and capillary ECs via Kir2 channels has been previously described as a mechanism to increase local cerebral blood flow (Longden and Nelson, 2015, Longden et al., 2017, Filosa et al., 2006, Haddy et al., 2006). Furthermore, a modeling study showcases the capillary EC Kir2 channel as a sensor of neuronal activity and highlights its impact on potassium – mediated neurovascular communication (Moshkforoush et al., 2020). […] Moreover, CPs are optimally positioned in the capillary bed (like cellular antennas of the EC-CP unit) to sense the microenvironment (Pfeiffer et al., 2021) and to amplify the hyperpolarization-mediated vascular response. Future studies using concurrent calcium imaging in ECs and CPs as well as cell-specific knock-out models are needed to gain more insights into the individual and the intercellular contribution of capillary ECs and CPs in potassium sensing and neurovascular coupling.”

OverallUsing predominantly an ex vivo acute brain slice preparation, the authors report differences in Ca^2+^ signaling signatures between EPs and CPs, lending additional experimental weight to the growing body of evidence that arteriole-proximate and more distal pericytes are functionally distinct. They describe differential effects of anesthesia on basal Ca^2+^ signaling in EPs (gradually eliminated) and CPs (largely unaffected) and show that Ca^2+^ signaling in EPs is reduced in brain slices. They further intimate (e.g., see Abstract, Figure 3 legend) that Ca^2+^ signaling in EPs requires blood flow; however, their data do not support this conclusion since flow effects on Ca^2+^ events were not tested directly. Importantly, the viability of cells in brain slices was not assessed following pharmacological manipulations that decreased Ca^2+^ signaling. In the absence of such control experiments, it is not possible to rule out tissue injury or improper slice maintenance as the cause of such decreases, especially given evidence for dead SMCs and EPs and unusually severe effects of U46619. Accordingly, some of the more compelling data from ex vivo pharmacological manipulations are suspect. The transient nature of EP Ca^2+^ responses recorded by 2-photon microscopy in cranial window model mice, reflecting signal dampening associated with the onset of anesthesia, similarly cast doubt on the significance of these in vivo findings. None of their data, including chemogenetic activation of hM3D(Gq)-DREADD-expressing neurons and pharmacological manipulations involving K^+^ and BaCl2, support the specific conclusion that neuronal K^+^ release causes the decrease/loss of pericyte calcium signaling. Importantly, pharmacological manipulations of Kir2 and KATP channels do not exclude endothelial cell contributions-especially given the importance of endothelial cell Kir2 in neurovascular coupling. More generally, the interconnected electrical properties of the microvasculature are not discussed or accounted for within the overall experimental design.

We appreciate the constructive feedback of the reviewer. We have now conducted a large battery of new experiments to address the concerns of (1) anesthesia by performing awake Ca^2+^ imaging and (2) cell viability (following manipulations that decreased Ca^2+^ signaling) by adding specific control measurements. We could also clarify the misleading notion of dead SMCs and EPs that appeared to be silent (low calcium activity) ex vivo in the absence of preconstruction. This resulted in several new figures which we presented and discussed specifically in the above responses. Moreover, we have discussed in the manuscript that we cannot exclude endothelial cell contributions and have now included an extended discussion on the interconnected electrical properties of the microvasculature, and the intriguing question whether pericytes could play a role in this complex cellular interplay.

Reviewer #2:This is an exciting and timely study to uncover the mechanisms and physiological relevance of pericyte calcium signaling. By using pericyte specific calcium sensing-reporter mice (Pdgfrbeta-CreERT2; GCaMP6s (Ai96-flox)), Gluck et al. identify distinct Ca signatures within the heterogeneous populations of pericytes along the brain vasculature including smooth muscle cells (SMCs), ensheathing pericytes (EPs; found along the branch orders 1-4), and capillary pericytes (CPs; found along > 4th order). While SMCs and EPs have synchronous calcium oscillations that are expectedly related to vessel diameter changes, CPs have asynchronous calcium signaling in somata and processes and there is no correlation with vessel diameter changes in the healthy anesthetized animal. This has been shown in some prior studies, but the current manuscript delves far deeper into the basis of pericyte calcium signals.To pharmacologically test how calcium signals are regulated in pericytes, they moved to ex vivo, brain slice experiments. They show that blocking VGCC and TRPC reduces CP calcium signals. Activating neurons using chemogenetics reduced CP calcium signals in vivo. KCl, to mimic neuronal activity, also reduced CP calcium and this effect is attenuated by blocking Kir2 and Kir6.1 ex vivo. Creating hypoxia-like conditions (in vivo: N2 or ex vivo: sodium azide) reduced calcium signals in CPs. Conversely, a number of vasoactive substances known to constrict arterioles lead to large calcium influxes in CPs, including ET-1, ATP and UDP-glucose.These studies pave new roads to understanding how pericytes are involved in NV coupling, and blood flow control. The approach is highly innovative and the in vivo-ex vivo approach is a major strength. However, there is a missing link that makes it difficult to understand the physiological relevance of the CP calcium signals as currently presented (comments 1 and 2). If addressed, this would substantially increase the impact of the work.1) It is not clear how calcium signaling in CPs is related to electrical conductance along the capillaries leading to arteriole diameter change. For example, how do CP calcium changes relate to calcium in SMCs and EPs (and/or arteriolar diameter) on the same vascular tree during NV coupling in vivo? How does CP calcium levels relate to endothelial activity? Is it possible to block CP K^+^ signaling in vivo to test the hypothesis that upstream arteriolar dilation will be decreased? Perhaps the relatively selective expression of KATP in CPs can be used to leverage an in vivo pharmacological experiment. It is not necessary to address all these questions, but some additional data to link CP calcium to arteriolar dilation would strengthen the paper and substantiate the compelling model in Figure 6.

We thank the reviewer for his positive and constructive feedback. We fully share the reviewer’s curiosity and are highly interested in how CP Ca^2+^ signals relate to NVC. Despite the complexity of the question, we performed additional experiments in vivo with an intent to address how neuronal activity-evoked changes in CP Ca^2+^ dynamics associate with SMCs and EPs and arteriolar diameters along the same vascular tree. We limited the neuronal DREADD (Gq) expression by AAV delivery close to the capillary bed, however, we cannot completely exclude partial neuronal expression in the vicinity of arterioles or arteries. When we activated neurons with clozapine, we observed a drop in Ca^2+^ activity in all mural cells along a connected vascular tree (see new Figure 5). This drop in Ca^2+^ activity was paralleled by a vessel diameter increase in all vascular compartments, i.e. artery, arteriole and capillary. Thus, we can say that the drop in CP Ca^2+^ activity associates with an increase in capillary diameter, however, it remains open whether this capillary diameter change is mediated by CPs or is an indirect effect of upstream artery/arteriole diameter changes. For now, we cannot answer how this relates to possible changes in endothelial activity (this would require e.g. simultaneous dual Ca^2+^ imaging in CPs and endothelial cells) and further investigations with different approaches are needed to address these questions. These could include sparse optogenetic stimulation of neurons close to CPs, and cell-specific conditional mutants (targeting e.g. Kir or KATP channels, and/or gap junctions) where the interaction between CPs or between CPs and endothelial cells could be studied. In light of recent findings showing how optogenetic activation of CPs may lead to capillary constriction, as shown by the reviewer’s lab (Hartmann et al., 2021), yet sensory stimulation did not evoke measurable capillary diameter changes (Hill et al., 2015) it may well be that a drop in CP Ca^2+^ activity (in response to certain levels of neuronal activity and increases in extracellular potassium) could be involved in facilitating capillary dilation. However, future studies are needed to fully resolve the role of CPs in neurovascular coupling. We have re-written the results and discussion to clarify these points and point out unresolved questions and future directions.

To further investigate mechanisms of signal transmission between CPs, we imagine an experiment (Author response image 1), under the hypothesis, that neuronal activity leads to a Ca^2+^ drop in CP (CP1) and that this Ca^2+^ drop is transmitted upstream along connected CPs (CP2, CP3) on a vascular branch. Using laser-ablation of a CP (CP2) that is in between connected CPs should possibly stop transmission of a Ca^2+^ drop. Further, the impact of gap-junction coupling between endothelial cells or pericytes and endothelial cells could be investigated by pharmacologically blocking gap-junction proteins or by cell-type specific knockouts.

**Author response image 1. sa2fig1:** Hypothetical experiment to investigate signal transmission between CPs along a vascular branch.

Furthermore, it is possible to perform dual Ca^2+^ imaging of endothelial cells and pericytes by either crossing acta2-RCaMP1.07 (JAX Stock.: 028345) and Cdh5BAC-GCaMP8 (JAX Stock.: 033342) or our here used cross between Pdgfrb-CreERT2 (JAX Stock.: 029684) and GCaMP6s (JAX Stock.: 028866) in combination with an endothelial specific AAV (Korbelin et al., 2016) for RCaMP1.07 sensor expression in capillary endothelial cells.

Ultimately, electrophysiological recordings of pericytes and endothelial cells, combined with selective pharmacology could provide more specific insights into the nature of signal spread (as for example showcased in a preprint by the group of Mark Nelson (Sancho et al., 2021)).

2) Local regulation of capillary flow: The presumption is that the sustained calcium changes evoked by vasomediators (ET-1, ATP) and neural activity are affecting local capillary diameter. This would be helpful to know because past studies have not been as rigorous in reporting vascular branch orders, leaving some ambiguity as the authors note. The data would help us understand whether CPs are autonomously able to regulate capillary diameter. This is important even if the stimulation becomes supra-physiological, as it clarifies whether CPs are a logical target for constriction in pathologies such as stroke, AD. Curiously, some of the data also show a slower component of calcium modulation can occur on the scale of tens of second to minutes (traces of Figure 5). Are these slow modulations related to capillary diameter? Similarly, how does N2-induced hypoxia result in vascular diameter changes along the various pericyte territories in relation to Ca signal changes?

We thank the reviewer for these insights.

Indeed, we could observe capillary diameter changes (constrictions) when brain slices were treated with vasomediators ET-1 or U46619 which evoked a large Ca^2+^ increase in CPs (Figure 3D and Figure 3 —figure supplement 2). However, we are reluctant to provide diameter measurements for these observations made in brain slices, since we only detected these vasculature changes through staining of stationary blood plasma, without actually seeing the vessel borders. To be more conclusive, repeating these experiments in future studies with a dual imaging approach using epifluorescent/transmitted light microscopy would yield more precise measurements of vessel diameters.

On the other hand, by increasing neuronal activity (Gq-DREADD activation) or transient hypoxia we could observe a drop in Ca^2+^ activity in CPs which associated with capillary diameter changes (dilation) in vivo (new Figure 5D, new Figure 6 —figure supplement 3). These measurements were acquired from timeaveraged (20 s) images of the vasculature.

Taking these two observations together, it is plausible that silencing and increasing CP Ca^2+^ activity could be involved in regulating capillary diameters. Our findings are in line with the recent report of how optogenetic stimulation of CPs induce capillary constrictions (Hartmann et al., 2021). How Ca^2+^ changes and cytoskeletal rearrangement in CPs regulate capillary diameter, and whether this occurs at the level of individual CPs or involves intercellular crosstalk with other mural cells and endothelial cells, needs to be determined in future studies.

3) Neural activity: DREADDs are gated by clozapine-N-oxide, rather than clozapine. It is correct that clozapine was used in this study, instead of its less bioactive counterpart? If so, a control non-DREADD expressing group might be needed to verify the intended goal of stimulating local neurons in vivo, as opposed to broad actions of clozapine. Further, it is suggested that basal neuronal activity in slice does not regulate CP calcium fluctuations because TTX has no effect. However, neuronal activity was not actually measured.

We thank the reviewer for his comment. We have chosen clozapine over clozapine-N-oxide (CNO) based on following reasons:

1) It has been shown, that clozapine is actually the effective molecule in DREADD activation, since CNO is back-converted to clozapine in vivo (Gomez et al., 2017).

2) Due to back-conversion of CNO to clozapine, CNO is in fact less bioactive than clozapine, requiring higher doses, which may in turn lead to unwanted side-effects (Jendryka et al., 2019).

We used clozapine at a dose of 30 µg/kg which is in the previously reported concentration range void of side-effects (Cho et al., 2020).

As discussed in an earlier response to reviewer 1 (see above, point 3 and 9), we showed that clozapine treatment itself had no direct effect on Ca^2+^ signaling in CPs when measured in an area outside of hSynGq-DREADD transduced neurons (new Figure 5 —figure supplement 1). Furthermore, we also show that the neuronal activation induced drop in Ca^2+^ in mural cells was not due to any lasting toxic side effects, since the same mural cells revealed normal Ca^2+^ activity when imaged on the following day (new Figure 5 —figure supplement 2).

Our findings of neuronal activity induced decrease of Ca^2+^ activity in cortical CPs are in line with a recent study showing how neuronal stimulation with odorants reduces CP Ca^2+^ levels in the olfactory bulb (Rungta et al., 2018).

It is indeed very intriguing that an increase in neuronal activity impacts CP Ca^2+^ dynamics. It could be that CP Ca^2+^ levels are regulated primarily by an acute rise in neuronal network activity (highly raising extracellular potassium concentrations). This could possibly explain why sensory whisker stimulation was reported to not influence CP Ca^2+^ dynamics (Hill et al., 2015).

Indeed, our TTX slice experiments revealed that basal neuronal activity (which is likely low in slices) has no major influence on CP Ca^2+^ dynamics (see Figure 6 —figure supplement 2). We have not measured neuronal activity during CP Ca^2+^ imaging, but we provide additional data showing that TTX silences action potential firing in our slice preparations (new Figure 6 —figure supplement 1). This may substantiate the notion that CPs primarily react (by decreasing Ca^2+^ activity) to acute changes of elevated neuronal activity. However, to define a specific level or type of neuronal activation requires future investigation.

4) In addition to a Non-DREADD expressing control, some additional controls would increase rigor. To ensure that the fluctuation in calcium dynamics in pericytes is not influenced by animal motion or passing RBCs in vivo, comparison to a general fluorescent reporter like GFP would be helpful. Vehicle control experiments are not described for the ex vivo pharmacological experiments. If these were performed, it would be good to show.

We agree with the reviewer, that control experiments are important to increase rigor.

As mentioned above, mural cell Ca^2+^ imaging was performed in a non-DREADD expressing brain region as internal control (new Figure 5 —figure supplement 1).

We have performed vehicle control experiments which are now added as a supplementary figure (new Figure 4 —figure supplement 1).

We are confident that the fluctuations in pericyte Ca^2+^ dynamics in vivo were not influenced by passing RBCs or motion given that CP Ca^2+^ signaling frequency were comparable to the recordings in acute brain slice conditions without any influence of passing RBCs and motion.

Reviewer #3:The Manuscript by Glück et al. examined the patterns of calcium transients in vascular mural cells in the live mouse brain and brain slices using two photon imaging of calcium sensor mice, in combination with pharmacological and chemogenetic manipulations. They found that pericytes exhibit distinct calcium dynamics compared to vascular smooth muscle cells (vSMCs). Pericytes in vivo have compartmentalized, irregular, higher-frequency calcium transients, which are not correlated with vessel diameter changes, whereas calcium fluctuation of vSMCs are highly correlated with changes in vessel diameter. In ex vivo preparations, pericytes retained their spontaneous calcium properties in contrast to its loss in SMCs. Combined with chemogenetics in vivo and pharmacological manipulation in slices, the authors demonstrated neuronal activity can decrease calcium response in pericytes by elevating extracellular potassium, which is mediated by Kir2.2, Kir6.1. Energetics state, and hypoxia challenge can also affect pericyte Ca^2+^ activation through KATP channel.Overall, this paper is technically rigorous and reproduces the majority of in vivo findings by recent publications (PMIDs: 26119027, 29937277). It also provides a more granular investigation of Ca^2+^ transients in pericytes (comparing soma and processes) and the pharmacological investigation of potassium channels and ATP in slices is also novel.

We thank the reviewer for his positive feedback to our work.

1) The main criticism we have about this paper is the nomenclature used to define the different mural cells which continues to confuse the field. Why use the title "signatures of calcium in brain pericytes" instead of simply saying in mural cells? This is surprising given that their data completely parallels data showing that SMCs and pericytes are distinct cells types. They introduce again the confusing term "ensheathing pericytes" despite their finding that EP essentially have identically characteristics to vSMCs (near identical calcium properties, contractility, expression of aSMA, circumferential anatomy, lack of compartmentalized Ca^2+^ signals and others) as well as findings from other groups including transcriptome data showing no such thing as subpopulations of pericytes (PMIDs: 29443965, 30129931) and data from uptake of small fluorescent molecules specifically by CP but not other mural cells (PMID: 28504673). We think it would be critical for this paper to clarify the field rather than continue to confuse the nomenclature and cite historical 100 old anatomical studies to justify it. Otherwise it is essential that they produce functional, structural and genetic evidence that there is such thing as a distinct ensheathing cell that is closer to pericytes than to vSMCs to justify the name. The logical terminology would be to call them "terminal vSMCs" or something similar to account for the minor morphological difference these cells have compared to the slightly more proximal vSMCs.

We thank the reviewer for this critical point and adapted the title of the work to refer to mural cells. We now also included data of spontaneous Ca^2+^ signals in SMCs (new Figure 2).

Regarding the term ensheathing pericyte (EP), we see the issue as being whether to name a cell after a function (vasomotor activity by smooth muscle actin) or morphology and earliest description of pericytes (Zimmermann, 1923). The apparently continuous transition in morphology of mural cells from SMCs to pericytes and a missing consensus on how to label different vascular segments further complicates the issue (Holm et al., 2018). In recent literature, vascular branches following an artery were called arteriole, precapillary arteriole or directly capillary. Classification of mural cell subpopulations along the vasculature according to branch orders (Grant et al., 2017) could help to make future studies more comparable between each other. We therefore chose to adapt this branch ordering system in our study as well. Although transcriptomic studies (Vanlandewijck et al., 2018, He et al., 2018) of mural cells suggest no subpopulation of pericytes, three different SMC clusters (arterial (a), arteriole (aa) and venular (v) SMCs) were found. However, the differentiation between aaSMCs and pericytes was based on the expression of smooth muscle actin (*Acta2*) which has been shown to stop abruptly at the transition from arteriole to capillary level. This approach directly defines cells on arterioles to be SMCs. Despite the fact that aaSMCs were found molecularly to be more closely related to aSMCs compared to capillary cells, these transcriptomic studies were not set to reveal more subtle changes between these cell clusters. Even more so, considering a relatively low abundance of EPs/aaSMCs compared to aSMCs or CPs in the reported dataset.

In terms of functional differences, a recent study by the Nelson lab found that these perivascular cells in question lack functional ryanodine receptors (RyR), which is a defining feature of smooth muscle cells (Gonzales et al., 2020).

Additionally, these perivascular cells (i.e. EP) indeed have a pericyte appearance with a protruding cell body and an approximately two-fold greater length along the longitudinal vascular axis (Berthiaume et al., 2021). We also think that the morphology speaks more to the site of pericytes. SMCs are ordered “rings” along the vessel with “sausage-like” nuclei (Author response image 2), while EPs show a protruding cell body with ovoid nuclei (Author response image 2i), which is also the case with CPs (Author response image 2).

**Author response image 2. sa2fig2:** (i) Confocal image of SMCs showing Merge, GCaMP6s (pdgfrβ-driven), and nuclear DAPI stain. (ii) Confocal image of an EP/terminal vSMCs showing Merge, GCaMP6s (pdgfrβ-driven), and nuclear DAPI stain. (iii) Confocal image of a CP showing Merge, GCaMP6s (pdgfrβdriven), and nuclear DAPI stain. Yellow arrows point to the respective cell-type. Scale bars: 10 µm.

Considering all of the above points, in our opinion, the jury is still out there whether to name the perivascular cell in question ensheathing pericyte or terminal vascular smooth muscle cell (or precapillary SMC). Therefore, we will state in the revised manuscript, that the term ensheathing pericyte (EP) is synonymously used with terminal vascular smooth muscle cell (vSMC). However, we are very open to discuss this further.

2) Although not essential for this project, it would have been nice to obtain concurrent endothelial and pericyte Ca recordings (using RCaMP and GCaMP sensors) to really understand the role of CP Ca^2+^. Is it similar to the described hyperpolarization propagation in endothelium? (PMID: 28319610), are there any interactions between CP and endothelium?

We agree with the reviewer that concurrent endothelial and pericyte Ca^2+^ recordings would be very helpful to understand the relationship between endothelial cells and CPs. However, this is beyond the scope of this work presented here and is an exciting question for a future study.

Several new studies hint to the importance of endothelial calcium signaling for adaptations of blood flow, shear forces and implications in pathologies, such as diabetes and hypertension (Thakore et al., 2021, Fancher and Levitan, 2020, Yamamoto et al., 2006).

Moreover, studies in retinal preparations could show an electrical signal transmission between adjacent CPs (Wu et al., 2006). Given the tight interactions of CPs and endothelial cells via peg-socket contacts (Ornelas et al., 2021) and possibly via gap junctions (Perrot et al., 2020), a similar hyperpolarization as in endothelial cells could also occur in CPs. Yet, further studies are needed to gain more insights into the signaling spread along these mural cells.

To address these open questions in the field we have added to the discussion the following sentences:

“However, Kir2 and KATP channels are also expressed in capillary ECs which mediate a hyperpolarization in response to increases in extracellular potassium (Longden et al., 2017). Given that there is evidence of EC and CP gapjunctional coupling in the retinal vasculature (Wu et al., 2006, Kovacs-Oller et al., 2020, Ivanova et al., 2017, Ivanova et al., 2019) and the lack of cell-type specificity of pharmacological manipulations, we cannot exclude that the observed potassium evoked calcium drop in CPs could be secondary to changes in capillary ECs. However, still very little is known of how capillary ECs interact and modulate CP functions or vice versa. Gap-junction coupling of capillary ECs and CPs would allow for electrical interconnection between these cells, forming a vascular relay (Ivanova et al., 2019). Activation of Kir2 and KATP channels likely induces cellular hyperpolarization in this EC-CP capillary unit, which could be transmitted between and within CPs and ECs through gap junctional coupling (Figure 7). Potassium sensing by CPs and ECs could potentiate the propagation of hyperpolarization from a site of elevated neuronal activity to upstream feeding arterioles.“

3) They should also comment on the fact that it is very difficult to target pericytes without having impact on other cell types by pharmacological manipulation. Only Kir6.6 (Kcnj8) blocking provides pericyte specificity, since BaCl2 can also affect endothelial Kir channels (PMID: 26840527), which may indirectly change pericytes calcium.

We agree with the reviewer and we have discussed in the revised manuscript the difficulty to interpret pharmacological manipulations regarding cell specificity (see also our response to comments of reviewer 1, point 10). We added several sentences (see also in the comment before) which discuss this issue.

4) Is there any directionality in the propagation of the Ca transients in pericytes in vivo?

This is a very interesting question. The way we recorded and quantified Ca^2+^ signals did not allow us to measure signal propagation. However, our impression is that there was no apparent signal directionality, yet this requires a more detailed analysis and employment of fast volumetric scanning of whole pericytes.

5) Do the microdomains of Ca in vitro become more synchronized between processes and Soma. If so, could this mean that in the in vitro prep the microenvironment around pericytes is more homogeneous than in vivo, thereby causing more homogeneity of Ca?

This is an interesting insight. We performed a cross correlation analysis to assess, whether process transients were co-occurring with somata transients. Comparing awake, anesthetized and ex vivo measurements revealed that process transients in awake mice were occurring on average 2 s before a soma transient (linear mixed model, post-hoc p = 0.025). However, this effect is skewed, since 99.5 % of all process transient peaks are in the same range (-10 to 10 s time to Soma (see histogram, middle panel of Author response image 3), grey dots represent outliers) as anesthetized and ex vivo measurements. There were no significant differences between anesthetized and ex vivo measurements (Author response image 3).

**Author response image 3. sa2fig3:** Correlation analysis of process and somata transients for ex vivo, anesthetized and awake Ca^2+^ – measurements. Grey points represent outlier values.

6) In their figure 6 model there is no mention of possible pericyte to pericyte or pericyte to SMC communication. This would be worth considering as these cells are likely to be gap junction coupled.

We thank the reviewer for pointing out this possibility. We updated the previous figure 6 (now new Figure 7) to include the possibility of pericyte to pericyte communication. See also our response to the last remark of Reviewer #1.